
# Constraining nucleation, condensation, and chemistry in oxidation flow reactors using size-distribution measurements and aerosol microphysical modelling

Anna L. Hodshire[1], Brett B. Palm[2,a], M. Lizabeth Alexander[3], Qijing Bian[1], Pedro Campuzano-Jost[2], Eben S. Cross[4,b], Douglas A. Day[2], Suzane S. de Sá[5], Alex B. Guenther[6,7], Armin Hansel[8], James F. Hunter[4], Werner Jud[8,c], Thomas Karl[9], Saewung Kim[6], Jesse H. Kroll[3,10], Jeong-Hoo Park[11,d], Zhe Peng[2], Roger Seco[6], James N. Smith[12], Jose L. Jimenez[2], Jeffrey R. Pierce[1]

[1]Department of Atmospheric Science, Colorado State University, Fort Collins, CO, 80523 USA
[2]Dept. of Chemistry and Cooperative Institute for Research in Environmental Sciences (CIRES); University of Colorado, Boulder, CO, 80309, USA
[3]Environmental and Molecular Sciences Laboratory, Pacific Northwest National Laboratory, Richland,WA, 99352, USA
[4]Department of Civil and Environmental Engineering, Massachusetts Institute of Technology, Cambridge, MA, 02139, USA
[5]School of Engineering and Applied Sciences, Harvard University, Cambridge, MA, 02138, USA
[6]Department of Earth System Science, University of California, Irvine, Irvine, CA, 92697, USA
[7]Division of Atmospheric Sciences & Global Change, Pacific Northwest National Laboratory, Richland,WA, 99352, USA
[8]Institute of Ion and Applied Physics, University of Innsbruck, Innsbruck, 6020, Austria
[9]Institute for Atmospheric and Cryospheric Sciences, University of Innsbruck, Innsbruck, 6020, Austria
[10]Department of Chemical Engineering, Massachusetts Institute of Technology, Cambridge, MA, 02139, USA
[11]National Center for Atmospheric Research, Boulder, CO, 80305, USA
[12]Department of Chemistry, University of California, Irvine, CA, 92697, USA
[a]Now at Department of Atmospheric Sciences, University of Washington, Seattle, WA, 98195, USA
[b]Now at Center for Aerosol and Cloud Chemistry, Aerodyne Research, Inc., Billerica, MA, 01821, USA
[c]Now at Institute of Biochemical Plant Pathology, Research Unit Environmental Simulation, Helmholtz Zentrum München,
Munich, 85764, Germany
[d]Now at Climate and Air Quality Research Department, National Institute of Environmental Research (NIER), Incheon, 22689, Republic of Korea

*Correspondence to*: Anna L. Hodshire (hodshire@rams.colostate.edu)

**Abstract.** Oxidation flow reactors (OFRs) allow the concentration of a given atmospheric oxidant to be increased beyond ambient levels in order to study secondary organic aerosol (SOA) formation and aging over varying periods of equivalent aging by that oxidant. Previous studies have used these reactors to determine the bulk OA mass and chemical evolution. To our knowledge, no OFR study has focused on the interpretation of the evolving aerosol size distributions. In this study, we
use size distribution measurements of the OFR and an aerosol microphysics model to learn about size-dependent processes in the OFR. Specifically, we use OFR exposures between 0.09-0.9 equivalent days of OH aging from the 2011 BEACHON-RoMBAS and the GoAmazon2014/5 field campaigns. We use simulations in the TOMAS (TwO-Moment Aerosol Sectional) microphysics box model to constrain the following parameters in the OFR: (1) the rate constant of gas-phase functionalization reactions of organic compounds with OH, (2) the rate constant of gas-phase fragmentation reactions of
organic compounds with OH, (3) the reactive uptake coefficient for heterogeneous fragmentation reactions with OH, (4) the



nucleation rate constants for three different nucleation schemes, and (5) an effective accommodation coefficient that accounts for possible particle diffusion limitations of particles larger than 60 nm in diameter.

We find the best model-to-measurement agreement when the accommodation coefficient of the larger particles ($D_p$>60 nm) was 0.1 or lower (with an accommodation coefficient of 1 for smaller particles), which suggests a diffusion limitation in the larger particles. When using these low accommodation-coefficient values, the model agrees with measurements when using a published $H_2SO_4$-organics nucleation mechanism and previously published values of rate constants for gas-phase oxidation reactions. Further, gas-phase fragmentation was found to have a significant impact upon the size distribution, and including fragmentation was necessary for accurately simulating the distributions in the OFR. The model was insensitive to the value of the reactive uptake coefficient on these aging timescales. Monoterpenes and isoprene could explain 24-95% of the observed change in total volume of aerosol in the OFR, with ambient semivolatile and intermediate-volatility organic compounds (S/IVOCs) appearing to explain the remainder of the change in total volume. These results provide support to the mass-based findings of previous OFR studies, give insight to important size-distribution dynamics in the OFR, and enable the design of future OFR studies focused on new particle formation and/or microphysical processes.

## 1 Introduction

Aerosols impact the climate directly, through absorbing and scattering incoming solar radiation (Charlson et al., 1992), and indirectly, through modifying cloud properties (Rosenfeld et al., 2008; Clement et al., 2009). Both of these effects are size-dependent, with larger particles dominating both effects. Particles with diameters ($D_p$) greater than 50-100 nm can act as cloud condensation nuclei (CCN) and particles with $D_p$ greater than 200-300 nm can absorb and scatter radiation more efficiently than smaller particles (Seinfeld and Pandis, 2006). The radiative forcing predictions of these effects remain amongst the largest uncertainties in climate modelling (Boucher et al., 2013), and thus climate predictions rely greatly upon accurate simulations or assumptions of the particle-size distributions. The majority of aerosol number globally is derived from photochemically driven new particle formation (NPF) of ~1 nm particles (e.g., Spracklen et al., 2008; Pierce and Adams, 2009a). These new particles are too small to impact climate, and they must grow through uptake of vapors and similarly sized particles while avoiding being lost by coagulation to larger particles in order to reach climatically relevant sizes (Westervelt et al., 2014). Thus, accurately simulating new particle formation and growth processes is a key step towards representing particle size distributions and predicting aerosol-climate effects in regional and global models that assess aerosol impacts. In the following paragraphs, we discuss the processes that shape new-particle formation and growth processes relevant to the analyses in this paper.

A large fraction of submicron aerosol mass is composed of organic aerosols (OA) (Murphy et al., 2006; Zhang et al., 2007; Jimenez et al., 2009; Shrivistava et al., 2017). OA is composed of thousands of often-unidentified compounds (Goldstein and



Galbally, 2007) and can be emitted directly in the particle phase as primary OA (POA) or formed as secondary OA (SOA) through gas-to-particle conversion. In SOA formation through the gas-phase, atmospheric oxidants (mainly OH, $O_3$, and $NO_3$) react with organic gases to form either less-volatile functionalized compounds or often more-volatile fragmentation products. If the oxidation products have a low-enough volatility, they may then partition to the particle phase, forming SOA

(Pankow et al., 1994; Donahue et al., 2006). The vapors may either partition to pre-existing particles or form new particles through NPF. Alternatively, the oxidation products could react in the particle phase to form lower volatility products that then remain in the particle phase (e.g., Paulot et al., 2009).

Controlled studies of SOA formation have traditionally used large reaction chambers with residence times of hours (often referred to as "smog chambers"). Chambers are susceptible to loss of both gases and particles to the walls of the chambers

(e.g., Krechmer et al., 2016; Bian et al., 2017). In order to enable the study of SOA formation from ambient air and limit wall losses, oxidation flow reactors (OFRs, i.e., the Potential Aerosol Mass [PAM] reactor; Kang et al., 2007, Lambe et al., 2011a) were developed to produce high and controllable oxidant concentrations and have short residence times (usually ~2 – 4 minutes), with the purpose of simulating hours to days or weeks of equivalent atmospheric aging (eq. days) in either laboratory or field experiments. Wall losses in OFRs can often be smaller than in large chambers due to shorter residence

times (e.g. Palm et al., 2016), although a direct comparison requires specification of the operating conditions, and losses in both types of reactors are still a subject of research. Studies with OFRs have shown SOA yields from precursor gases are similar to yields from smog chambers (Kang et al., 2007; Lambe et al., 2011b, 2015; Palm et al., 2018). Previous field studies with OFRs have focused on bulk aerosol mass formation and aging, and bulk chemical evolution (e.g., Ortega et al., 2013, 2016; Tkacik et al. 2014; Palm et al. 2016, 2017a, 2018). Ortega et al. (2016) and Palm et al. (2016) showed that size

distributions in OFR output were dynamic as a function of time and aging. However, to the best of our knowledge, no ambient OFR study has focused on the aerosol size distributions that form and evolve within the OFR. Processes that could help shape the size distribution within the OFR are the same as those that take place in the real atmosphere, and include nucleation, condensation of vapors, coagulation, the rate of gas-phase oxidation with OH, gas-phase fragmentation with OH, vapor-uptake and/or particle diffusion limitations, reactive uptake growth mechanisms including accretion reactions and

acid-base reactions, heterogeneous reactions, and wall losses of both vapors and particles. Many of these processes have uncertainties associated with them, necessitating model-to-measurement comparisons and sensitivity studies. Using an OFR extends the parameter space over which comparisons can be made, compared to using only ambient data where parameter variations are narrower.

Nucleation, i.e., the formation of new ~1 nm particles, can involve a number of species, including water, sulfuric acid,

ammonia, amines, ions, and certain low-volatility organic compounds (e.g., Kulmala et al., 1998; Vehkamaki et al., 2002; Kulmala et al., 2002; Napari et al., 2002; Laakso et al., 2002; Yu et al., 2006a; Yu et al., 2006b; Metzger et al., 2010; Almeida et al., 2013; Jen et al., 2014; Riccobono et al., 2014). Along with multiple species, observations indicate that



numerous physical and chemical reactions can be involved (e.g., Zhang et al., 2004; Chen et al., 2012; Almeida et al., 2013; Riccobono et al., 2014). Recent studies have pointed to the importance of nucleation involving sulfuric acid and oxygenated organic compounds over the forested continental boundary layer (BL) (e.g., Metzger et al., 2010; Riccobono et al., 2014). However, controlled nucleation and growth studies in smog chambers or oxidation flow reactors involving organics have

traditionally focused on organics formed from the oxidation of a single precursor vapor, such as $\alpha$-pinene. Previous chamber studies have examined NPF from plant emissions (e.g., Joutsensaari et al., 2005; Vanreken et al., 2006), but to our knowledge no studies have systematically investigated nucleation and growth mechanisms in OFR or other types of reactors using ambient air as the precursor source.

Condensation of vapors to newly formed aerosol particles as well as pre-existing particles increases the total aerosol particle

mass, but the net condensation rate to differently sized particles is dependent upon the volatility of the vapors. The lowest-volatility vapors condense essentially irreversibly onto particles of all sizes (i.e. "kinetically limited" or irreversible condensation; Riipinen et al., 2011, Zhang et al., 2012). Semi-volatile vapors (with non-trivial partitioning fractions in both the particle and gas phases at equilibrium) have a net condensation to particles that is determined by reversible partitioning (i.e. quasi-equilibrium condensation; Riipinen et al., 2011, Zhang et al., 2012). Kinetically limited condensation is gas-

phase-diffusion limited and only possible for compounds with effective saturation concentrations ($C^*$; Donahue et al., 2006) $< \sim 10^{-3}$ $\mu$g m$^{-3}$ (e.g., low and extremely low volatility organic compounds; LVOCs and ELVOCs); the net SOA uptake to a particle is proportional to the Fuchs-corrected surface area of the particle (Pierce et al., 2011). Conversely, thermodynamic condensation primarily involves semi-volatile organic compounds (SVOCs) with $C^* \sim 10^{-1}$ –$10^2$ $\mu$g m$^{-3}$ that quickly reach equilibrium between the gas and particle phases for all particle sizes; as a result, the net SOA uptake to a particle is

proportional to the organic mass (or volume) of the particle (Pierce et al., 2011).

The gas-phase oxidation rates of organic vapors as well as the competition between gas-phase functionalization (the addition of polar, oxygen-containing functional groups, generally lowering the volatility of the species) and gas-phase fragmentation (the cleavage of C-C bonds, with each reaction typically creating two higher-volatility products) influences the changes in volatilities of organic species from atmospheric oxidation (e.g.,, Kroll et al., 2009). Gas-phase oxidation rates have been

well-quantified for many individual species in the lab (e.g., Atkinson and Arey, 2003a), but less is known about gas-phase oxidation rates that may be appropriate for lumped organic vapors in ambient air. Generally, a representative reaction rate constant ($k_{OH}$) for a given oxidant is chosen to describe oxidation of organic species present in ambient air in modelling studies that may be a function of organic-vapor volatility (e.g., Jathar et al., 2014; Bian et al., 2017). Beyond $k_{OH}$ values, the volatility of the reaction products is also important. Recent modelling studies have shown significant impacts to the SOA

budget when fragmentation reactions were included relative to the assumption that all products were purely functionalized (e.g., Shrivistava et al., 2013; 2014; 2016). Several recent laboratory studies point to the likely increasing importance of



fragmentation reactions as organic vapors age and become more functionalized (Jimenez et al., 2009; Kroll et al., 2009, 2011; Chacon-Madrid et al., 2010; Chacon-Madrid and Donahue, 2011; Lambe et al., 2012; Wilson et al., 2012). Reduced organic vapors generally functionalize without fragmentation upon oxidation, decreasing their volatility. However, the probability of fragmentation (and an increase in overall volatility) increases after repeated oxidation reactions (if the

molecule does not leave the vapor phase first). Hence, in addition to decreasing the overall mass yield of SOA, gas-phase fragmentation reactions reduce the production of the lowest volatility species that condense through the gas-phase-diffusion limited pathway and thus the balance between fragmentation reactions and purely functionalization reactions may impact the size-dependent condensation of SOA in addition to the overall SOA yield. However, the balance between gas-phase functionalization reactions and fragmentation reactions are not well constrained for ambient organic mixtures.

Particle-phase reactions also shape OA mass and the size distribution. Heterogeneous reactions between OH and organics at the surface of the particle can yield fragmentation products with high-enough volatilities to evaporate from the particle (e.g., Kroll et al., 2009), resulting in particle mass loss. Heterogeneous reactions contribute to aerosol aging and influence aerosol lifetime (George and Abbatt, 2010; George et al., 2015; Kroll et al., 2015). Many laboratory studies have reported uptake coefficients of OH, $\gamma_{OH}$, defined as the fraction of OH collisions with a particle-phase compound that result in a reaction,

with values of effective $\gamma_{OH}$ ranging from $\leq 0.01$ to $> 1$, depending upon the reaction conditions (e.g., McNeill et al., 2008; Park et al., 2008; George and Abbatt, 2010; Liu et al., 2012; Slade and Knopf, 2013; Arangio et al., 2015; Hu et al., 2016). This heterogeneous OA loss pathway is important in OFRs at very high OH concentrations (corresponding to exposures of $\gg 1$ day) (e.g., Ortega et al., 2016; Hu et al., 2016; Palm et al., 2016), and $\gamma_{OH} \sim 0.6$ has been measured for ambient OA (Hu et al., 2016). Conversely, particle-phase reactions including acid-base and accretion reactions can contribute to particle mass

through the formation of lower-volatility products than the parent molecules (e.g., Pankow 2003; Barsanti and Pankow 2004; Pinder et al., 2007; Pun and Seigneur et al., 2007).

SOA uptake rates may be limited by the phase state of SOA through particle diffusion limitations. Traditionally, SOA was viewed as a liquid mixture; however, SOA have been observed in solid and amorphous phases in both laboratory and field studies (Virtanen et al., 2010; 2011). Measurements taken in 2013 and during the GoAmazon2014/5 campaign (Martin et al.,

2016; 2017) found that SOA produced from oxidation products from the Amazonian rainforest tended to be primarily liquid whereas SOA influenced by anthropogenic emissions (both from the Manaus pollution plume and biomass burning) tended to have higher fractions of semisolid and solid aerosol (Bateman et al., 2015; 2017). Mixing in these solid or amorphous phases could decrease (Cappa et al., 2011; Vaden et al., 2011), leading to decreases in gas-particle partitioning rates (Shiraiwa and Seinfeld, 2012). The impacts of the changes in phase state from liquid to solid/amorphous matters less for

SOA uptake at smaller particle sizes ($D_p < \sim 100$ nm), but increases more with increasing particle sizes (Shiraiwa et al., 2011). Hence, one may hypothesize that vapor-uptake limitations may favor the uptake of organics to smaller particles relative to when particles are liquid and do not have vapor-uptake limitations. This boost of growth to the smallest particles



due to vapor-uptake limitations may be strong if coupled with particle-phase oligomerization reactions (Zaveri et al. 2014). Zaveri et al. (2017) found that in order to model the growth of bimodal aerosol populations formed from either isoprene or α-pinene and isoprene oxidation products, the intraparticle bulk diffusivity of the accumulation mode had to be slower (an order of magnitude less) than that of the diffusivity of the Aitken mode. Yatavelli et al. (2014) showed that gases and

particles appeared to be in equilibrium over a timescale of 1 hr at the BEACHON-RoMBAS site; however, OFR timescales are significantly shorter. Recent parameterizations for α-pinene SOA, an important compound at the BEACHON-RoMBAS site, are inconclusive about the diffusion timescale of these particles due to limitations in the input data (Maclean et al., 2017).

Each of the processes discussed above (nucleation, condensation of vapors, gas-phase functionalization and fragmentation

reactions, heterogeneous reactions, accretion reactions, acid-base reactions, and particle diffusion limitations) could have very different timescales in the OFR as compared to the ambient atmosphere; for example, the chemistry timescale will typically be much shorter than the condensation and coagulation timescales in the OFR since the OFR OH concentrations can greatly exceed that of the ambient OH concentrations. Thus, models must be used to help interpret the OFR processes to determine how the observations relate to the ambient atmosphere. In this study, we use OFR measurements taken from two

field locations. In the first, an OFR was deployed during the BEACHON-RoMBAS field campaign (Ortega et al., 2014) that took place in a montane ponderosa pine forest in Colorado, USA during July-August 2011. The second is the GoAmazon2014/5 field campaign (Martin et al., 2016; 2017) that occurred from January 2014-December 2015 in the State of Amazonia, Brazil, in the central Amazon basin. OFR data from each of these two campaigns have been analyzed in previous work (Palm et al. 2016; 2017a; 2018; Hunter et al., 2017) to understand the bulk OA mass and chemical evolution

in the OFR. These analyses showed that the presence of unspeciated S/IVOCs contribute substantial OA mass production in the OFR at both locations. However, previous work has not analyzed the evolving aerosol size distribution in the OFR to gain insight into nucleation and growth processes. In this paper, we extend the analysis of these ambient datasets using the measured aerosol size distributions and a model of aerosol microphysics in the OFR.

## 2 Methods

### 2.1 OFR method

The aerosol measurements investigated in this work were of ambient air before and after oxidation in a Potential Aerosol Mass (PAM) reactor, which is a type of OFR (Kang 2007, Lambe 2011a). This OFR is a cylindrical aluminium tube with a volume of 13 L and a typical residence time of 2–4 min. OH radicals were produced inside the OFR by photolysis of

ambient $H_2O$ and concurrently produced $O_3$ using 185 and 254 nm emissions from low pressure mercury UV lamps. The OH concentrations in the OFR were stepped over a range from $\sim 8 \times 10^7$ to $9 \times 10^9$ molec cm$^{-3}$ by adjusting the UV lamp photon



flux, with only data near the lower end of the range investigated in this work (see Table 2). The OFR was operated outside of the measurement trailer under ambient temperature and humidity (but protected from direct sun). This allowed avoiding the use of an inlet, which minimized any possible losses of semivolatile or sticky SOA precursor gases to inlet walls. Further OFR sampling and measurement details for the data used in this work can be found in Palm et al. (2016, 2017, 2018). The

chemical regime was relevant to ambient OH oxidation, as discussed in detail in Peng et al. (2015, 2016).

### 2.2 Field campaigns

### 2.2.1 BEACHON-RoMBAS campaign

The BEACHON-RoMBAS field campaign (referred to as BEACHON hereafter) took place in July–August 2011 at the Manitou Experimental Forest Observatory near Woodland Park, Colorado (Ortega et al., 2014). The sampling site, located in

a ponderosa pine forest in a mountain valley, was influenced mainly by 2-methyl-3-buten-2-ol (MBO) during the day and monoterpenes (MT) at night. During BEACHON, an OFR was used to measure the amount and properties of SOA formed from the oxidation of real ambient SOA precursor gases and ambient aerosol. Ambient particles and SOA formation after OH oxidation in the OFR (and also $O_3$ or $NO_3$-only oxidations (Palm et al., 2017), which are not investigated in this work) were sampled using an Aerodyne high-resolution aerosol mass spectrometer (HR-ToF-AMS, hereafter referred to as AMS)

and a TSI Scanning Mobility Particle Sizer (SMPS). Details of OFR sampling can be found in Palm et al. (2016; 2017; 2018). Ambient $SO_2$ concentrations were measured using a Thermo Environmental Model 43C-TLE analyzer. VOC concentrations were quantified using a high-resolution proton-transfer reaction time of flight mass spectrometer (PTR-TOF-MS; Graus et al. 2010; Kaser et al. 2013). Ensemble mass concentration of ambient S/IVOCs in the range of $C^*$ from $10^1$ to $10^7$ µg m$^{-3}$ were measured using a novel thermal-desorption electron impact mass spectrometer (TD-EIMS; Cross et al. 2013;

Hunter et al. 2017). More details pertaining to the use of these instruments in measuring SOA formation in the OFR can be found in Palm et al. (2016).

### 2.2.2 GoAmazon2014/5 campaign

The GoAmazon2014/5 field campaign (referred to as GoAmazon hereafter) took place in the area surrounding Manaus, Brazil, in central Amazonia (Martin et al., 2016; 2017), investigating the complex interactions between urban, biomass

burning, and biogenic emissions. OFR measurements of SOA formation from OH oxidation of ambient air (and also $O_3$-only oxidation, not investigated here) were taken at the "T3" site downwind of Manaus during two intensive operating periods (IOP1 during the wet season and IOP2 in the dry season) to study the contributions of the various emission sources to potential SOA formation. The dry season results were chosen for investigation in this study due to the generally larger concentrations of gases, particles, and potential SOA formation than during the wet season. Whereas SOA formation at the

BEACHON site was dominated by a single source type (biogenic gases, related to MT), the "T3" site was influenced by a complex mixture of biogenic and anthropogenic emissions (Martin et al. 2016, Palm et al. 2018). Again, ambient particles



and SOA formation after OH oxidation in the OFR were sampled by an AMS and an SMPS. Ambient $SO_2$ concentrations were sampled using a Thermo Fisher Model 43i-TLE $SO_2$ Analyzer. Ambient VOCs were sampled using a PTR-TOF-MS. More details pertaining to the use of these instruments in measuring SOA formation in the OFR can be found in Palm et al. (2018).

**2.3 TOMAS-VBS box model**

**2.3.1 Model description**

In this study, we use the TwO-Moment Aerosol Sectional (TOMAS) microphysics zero-dimensional (box) model (Adams and Seinfeld, 2002; Pierce and Adams, 2009b; Pierce et al., 2011) combined with the Volatility Basis Set (VBS; Donahue et al., 2006) as described in Bian et al. (2017). This version of TOMAS-VBS simulates condensation, coagulation, and
nucleation, and it has a simple organic vapor aging scheme that moves an organic species down in volatility upon reaction with an OH molecule (Bian et al., 2017). The simulated aerosol species are sulfate, organics, and water within 40 logarithmically spaced size sections from 1.5 nm to 10 $\mu$m. We simulate 6 organic "species" within the VBS, representing lumped organics with logarithmically spaced effective saturation concentrations ($C^*$) spanning $10^{-4}$ to $10^6$ $\mu$g m$^{-3}$ (spaced apart by factors of 100). The $C^*=10^{-4}$ $\mu$g m$^{-3}$ bin represents extremely-low-volatility organic compounds (ELVOCs), the
$C^*=10^{-2}$ $\mu$g m$^{-3}$ bin represents low-volatility organic compounds (LVOCs), the $C^*=10^0$ $\mu$g m$^{-3}$ and $C^*=10^2$ $\mu$g m$^{-3}$ bins represents semivolatile organic compounds (SVOCs), and the $C^*=10^4$ $\mu$g m$^{-3}$ and $C^*=10^6$ $\mu$g m$^{-3}$ bins represents intermediate-volatility organic compounds (IVOCs), following the conventions proposed by Murphy et al. (2014). In the rest of this section, we discuss the base model setup and assumptions. In Section 2.3.3, we discuss the uncertainty space that we test in this study.

In this study, gas-phase functionalization is modelled by assuming that the organic compounds within the VBS bins react with OH and products from this reaction drop by one volatility bin (a factor of 100 drop in volatility). As a base assumption of the rate constants of our vapors in the VBS bins reacting with OH ($k_{OH}$), we use the relationship developed for aromatics by Jathar et al. (2014), based on data from Atkinson and Arey (2003a):

$$k_{OH} = -5.7\times10^{-12}ln(C^*) + 1.14\times10^{-10} . \tag{1}$$

As the assumption that the ambient mixture of S/IVOCs is similar to those of aromatics may not be suitable, we treat the rate constants for this volatility-reactivity relationship as an uncertain parameter that we vary in this study (Section 2.3.3).





We account for gas-phase fragmentation reactions separately by allowing one OH reaction with a molecule in the lowest volatility bin ($C^*=10^{-4}$ $\mu$g m$^{-3}$; assumed to be an ELVOC molecule) to lead to an irreversible fragmentation into non-condensable volatile products that are no longer tracked in the model. Realistically, fragmentation reactions occur for vapors across the whole range of volatilities; however, the likelihood of fragmentation increases with increasing levels of oxidation

(Kroll et al., 2011) and an increase in oxidation is often correlated with a decrease in volatility (Donahue et al., 2006; Kroll et al., 2011). We only allow for fragmentation of species in our lowest volatility bin in order to limit the number of parameters in our study, but we acknowledge that this is a limitation of this study. We discuss the potential implications of only allowing fragmentation in the lowest volatility bin in the conclusion section. Our base assumption for this rate constant is $10^{-10}$ cm$^3$ s$^{-1}$.

We further account for monoterpenes (MT) oxidation by OH for both campaigns and isoprene oxidation by OH for GoAmazon in the model. Palm et al. (2016) determined that on average during the BEACHON campaign, MT contributed 20% of the measured SOA formation (87% of the SOA predicted from all measured traditional VOC precursors), with sesquiterpenes (SQT), isoprene, and toluene contributing an additional 3% of the measured SOA formation (and the remaining 13% of predicted SOA from VOCs). Since these other VOCs contributed a minor amount to the measured SOA

formation, they were not included in this analysis. S/IVOCs at BEACHON contributed the remaining 77% towards SOA formation, and were likely the main source for new particles in the OFR. It was observed that for the GoAmazon campaign during the dry season, the approximate average contribution to the measured SOA was 4% from isoprene and 4% from MT (comprising a combined 50% of predicted SOA mass from measured VOCs), with an 8% remaining contribution towards the measured SOA (and remaining 50% of predicted SOA) coming from SQT, benzene, toluene, xylenes, and trimethylbenzene

(TMB), combined. Thus, less of the total predicted SOA can be described by the VOCs included in the model (isoprene and MT) for the GoAmazon simulations than can be described for the BEACHON campaign. The remaining 83% of measured SOA formation was found to have come from unmeasured S/IVOCs, so again S/IVOCs were likely the main source for new particles in the OFR. Including the other VOCs would only increase the model-predicted SOA yield from the initial VOCs by a few tenths of a $\mu$g m$^3$, and decrease the model-predicted SOA yield from the initial S/IVOCs by a similar amount, and

so they were excluded for simplicity.

The products of both MTs and isoprene oxidation enter the model's volatility bins in the vapor phase. For MT SOA production, we use the product yields for $\alpha$-pinene OH oxidation chamber experiments of Henry et al. (2012) for the $C^*=10^{-2}$ to $C^*=10^4$ $\mu$g m$^{-3}$ bins and the average OH oxidation yield for ELVOCs from four different terpene species of Jokinen et al. (2014) for the $C^*=10^{-4}$ $\mu$g m$^{-3}$ bin (Table 1). However, the wall loss correction applied in Henry et al. (2012) may not be

appropriate (Zhang et al., 2014) and hence these yields may contribute an additional source of uncertainty that we do not




explore in this paper. The isoprene SOA yields (Table 1) are for low $NO_x$ conditions (Tsimpidi et al. 2010), with the OH oxidation yield of isoprene from Jokinen et al. (2014) for the $C^*=10^{-4}$ $\mu$g m$^{-3}$ bin. In the OFR under OH oxidation, $NO_x$ is rapidly oxidized to $HNO_3$ (Li et al., 2015; Peng and Jimenez, 2017), and thus the assumption of using SOA yields developed under low $NO_x$ conditions are valid for the OFR exposures taken during BEACHON and GoAmazon. We use the rate

constants of OH oxidation for MT and isoprene of $5\times10^{-11}$ cm$^3$ molec$^{-1}$ s$^{-1}$ and $1\times10^{-10}$ cm$^3$ molec$^{-1}$ s$^{-1}$, respectively (Atkinson and Arey, 2003a). In this study, TOMAS-VBS does not track the MT and isoprene oxidation products once they enter the VBS scheme separately from the products of other precursors, and further oxidation of these products follows the $k_{OH}$ assumptions above. Although this assumption may be reasonable for MTs, studies in isoprene-dominated forests have shown that NPF appears to be suppressed in the regions studied even when monoterpene emissions are sufficiently high (Bae et al.,

2010; Kanawade et al., 2011; Pillai et al., 2013; Haller et al., 2015; Yu et al., 2015; Lee et al., 2016). Hence, the products of isoprene oxidation likely do not age similar to monoterpenes (e.g., Krechmer et al., 2015), but we do not account for this possible effect in our model.

We simulate heterogeneous fragmentation reactions of particle-phase organics in all VBS bins by OH. The resulting particle mass loss is modeled in TOMAS through:

$$\frac{dM_K\,[K,J]}{dt} = \gamma_{OH} J_{OH} \frac{M_K[K,J]}{\Sigma M_K[K,J]} \frac{MW_{loss}}{N_a} \quad , \tag{2}$$

where $M_K$ indicates the mass in a size section, $K$ and $J$ indicate the size bin and particle-phase species, $J_{OH}$ is the rate of molecules of OH hitting a particle, $MW_{loss}$ is the mass lost per reaction (taken here to be 250 amu; Hu et al., 2016), respectively, $N_a$ is Avogadro's number, and $\gamma_{OH}$ is the reactive uptake coefficient for heterogeneous reactions with OH. Our base value of $\gamma_{OH}$ is 0.6, following the measurements of Hu et al. (2016) in a very similar OFR field experiment, but we treat

$\gamma_{OH}$ as an uncertain parameter that we vary in this study (Section 2.3.3).

In this work, we explore three different possible nucleation schemes. The first two use a $H_2SO_4$-organics nucleation mechanism, using the nucleation parameterization of Riccobono et al. (2014),

$$J_{ORG} = k_{NUC}[H_2SO_4]^p[BioOxOrg]^q \,, \tag{3}$$

where $k_{NUC}$ is the nucleation rate constant, BioOxOrg represents later-generation oxidation products of biogenic

monoterpenes, and the exponents $p$ and $q$ represent the power law dependence of $J$ upon the concentrations of sulfuric acid and BioOxOrg. In Riccobono et al. (2014), $J_{ORG}$ was parameterized for the mobility diameter of 1.7 nm; in TOMAS, the





median dry diameter of the smallest bin is 1.2 nm. In this study, we use the ELVOC ($C^*=10^{-4}$ $\mu$g m$^{-3}$) bin of the TOMAS VBS scheme to represent the BioOxOrg concentration;

$$J_{ORG} = k_{NUC}[H_2SO_4]^p[ELVOC]^q .$$
$$(4)$$

Our primary nucleation scheme, referred to here as NUC1, uses the values of $p = 2$, $q = 1$, and a base value of $k_{NUC} = 1\times10^{-21}$ cm$^6$ molec$^{-1}$ s$^{-1}$. We will refer to this $k_{NUC}$ as $k_{NUC1}$ for the remainder of the manuscript. For comparison, for $p = 2$ and $q = 1$, Riccobono et al. (2014) found a $k_{NUC1}$ value of $3.27\times10^{-21}$ cm$^6$ molec$^{-1}$ s$^{-1}$ at 278 K. We acknowledge that the values of $p$ and $q$ are also uncertain (Riccobono et al., 2014) and we do a further sensitivity study for the nucleation parameterization, referred to here as NUC2, using $p = 1$, $q = 1$, and a base value of $k_{NUC2} = 5\times10^{-13}$ cm$^3$ molec$^{-1}$ s$^{-1}$. NUC2 can be thought to

account for possible saturation effects that could occur in the OFR that would result in shallower slopes (p and q) (Almeida et al. 2013; Riccobono et al., 2014). For comparison, Metzger et al. (2010) found a value of $k_{NUC2} = 7.5 \pm 0.3 \times10^{-14}$ cm$^3$ molec$^{-1}$ s$^{-1}$ (temperature not reported) when they constrained $p$ and $q$ to be both one. However, their study used the lowest-volatility oxidation products of 1,3,5-trimethylbenzene as the BioOxOrg proxy (Eq. 4), which is an anthropogenic SOA precursor. Although a temperature-dependent form of Eq. 4 has been developed (Yu et al., 2017), we instead here are fitting

the nucleation rate constant to the temperature of the measurements (Table 2). For each of these nucleation schemes, we treat $k_{NUC}$ as an uncertain parameter that we vary in this study (Section 2.3.3.).

We further explore the possibility of a sulfuric-acid only nucleation scheme, as some nucleation schemes used in models only rely upon the concentration of sulfuric acid (e.g., Spracklen et al., 2008, 2010; Westervelt et al., 2014; Merikanto et al., 2016) by using an activation nucleation scheme (Kulmala et al., 2006) for our third nucleation scheme, referred to here as

ACT, in which existing clusters are activated:

$$J_{ACT} = A[H_2SO_4] ,$$
$$(5)$$

where $A$ is referred to as the activation coefficient. Previous studies of activation nucleation have found fits for $A$ of between $3.3\times10^{-8}$ and $6\times10^{-6}$ s$^{-1}$ for a boreal forest (Sihto et al., 2006; Riipinen et al., 2007) and between $2.6\times10^{-6}$ and $3.5 \times10^{-4}$ s$^{-1}$ for a polluted environment (Riipinen et al., 2007). We use as a base $A$ value $2\times10^{-6}$ but treat this as an uncertain parameter

(Section 2.3.3.).





We include a simple approximation of potential vapor-uptake and/or particle diffusion limitations by setting an adjustable accommodation coefficient ($a_{EFF}$) that is fixed to 1 for particles below 60 nm in diameter but can vary between 0.01 and 1 for particles above 60 nm in diameter (see Section 2.3.3. for further discussion). This simple scheme allows the uptake of OA vapors to larger particles to be slowed relative to the uptake to smaller particles, due to the longer diffusion timescales in the larger particles (Shiraiwa et al., 2011). The cutoff of 60 nm was chosen because upon initial inspection of simulations with the accommodation coefficient set to 1 for all particle sizes, it was seen that the growing new aerosol in the Aitken mode (particles largely below 60 nm) did not require any slowing of growth but the aerosol in the accumulation mode (particles largely above 60 nm) did require slowing of growth. We acknowledge that our method here is a crude approximation of particle diffusion limitations. However, with only very limited knowledge of particle-phase diffusivities and how they may vary with size (Zaveri et al. 2017), composition, and/or ambient conditions, such as temperature and relative humidity, we use this simple scheme as a way of determining if vapor-uptake limitations, potentially due to particle-phase-diffusion limitations, may be important in limiting the growth of larger particles relative to the smallest particles.

In this study, we do not simulate acid-base reactions and accretion reactions. No gas-phase bases (ammonia or amines) were measured during either campaign, making modelling acid-base reactions in TOMAS too unconstrained. Further, the model simulations point towards high concentrations of ELVOCs in the gas-phase needed to facilitate nucleation (Section 3.1), indicating that gas-phase ELVOC production may be the dominant ELVOC-formation pathway over particle-phase ELVOC production (through accretion reactions and/or acid-base reactions). However, we cannot rule out ELVOC production in the particle phase through particle-phase reactions, as ELVOCs are in the particle phase at equilibrium.

We simulate loss of low-volatility vapors to the OFR walls using a first-order rate constant, $k_{wall}$=0.0025 s$^{-1}$, estimated in Palm et al. (2016) following McMurry and Grosjean (1985). Palm et al. (2016) estimated this loss for condensable (low-volatility) species; we extend this loss to the $C^*$=10$^{-2}$ $\mu$g m$^{-3}$ (LVOC) and the $C^*$=10$^{-4}$ $\mu$g m$^{-3}$ (ELVOC) bins in our VBS system. We use this value of $k_{wall}$ for both the BEACHON and the GoAmazon OFR simulations. We assume that the wall losses for higher volatility species and particles are slow and ignore them (this was verified for particles by Palm et al., 2016).

For the BEACHON simulations, we use the residence time distribution (RTD) in the OFR of Palm et al. (2017) assuming non-Brownian motion (their Figure S1). The RTD is less-well characterized for GoAmazon; we use the RTD for particles from Lambe et al. (2011a), but as discussed in Palm et al. (2018), the RTD from Lambe et al. (2011a) is likely more skewed than for the OFR used at GoAmazon, due to the larger inlet at GoAmazon. The SMPS data for both campaigns were corrected for diffusion losses to the walls of the sampling lines (Palm et al., 2016; Palm et al., 2018).



We simulate coagulation using the Brownian kernel in Seinfeld and Pandis (2006). However, we do not expect coagulation to be a dominant process in our OFR simulations. The condensation sink timescale for the measured size distributions were on the order of 0.5-5 minutes, which corresponds to coagulation sink timescales on the order of 1-10 minutes for 1 nm particles, 2.5-25 minutes for 2 nm particles, and 5-50 minutes for 3 nm particles (Dal Maso et al., 2002). Thus, in some cases

the coagulation sink timescales for the freshly nucleated particles were similar to the residence time. However, in most cases, freshly nucleated particles grew to at least 20 nm within the OFR, so the nucleated particles spend only a small fraction (<10%) of the residence time at sizes smaller than 3 nm. Hence, the coagulation timescale of the growing particles is overall much longer than the residence time, and we expect on the order of 10% or fewer of the nucleated particles to be lost by coagulation in these OFR experiments.

**2.3.2 Model inputs**

Inputs to TOMAS to initialize each OFR exposure simulated from the BEACHON and GoAmazon field campaigns are given in Table 2; each input represents the initial condition present at the start of the exposure. The initial ambient size distribution from each campaign's SMPS is also used (Figs. 1 and S1, black lines). The initial S/IVOC concentration (as measured by the TD-EIMS) is evenly divided between the $C*=10^2$ to $C*=10^6$ $\mu$g m$^{-3}$ bins in TOMAS. Although The TD-

EIMS reported ambient concentrations decadally between $C*=10^1$ to $C*=10^7$ $\mu$g m$^{-3}$, differences in mass concentrations per bin were small (Palm et al, 2016; Hunter et al, 2017) and thus our assumed division should be within experimental uncertainty. The initial total aerosol mass (as measured by the AMS) is evenly divided between the $C*=10^{-4}$ to $C*=10^{-2}$ $\mu$g m$^{-3}$ bins, consistent with the overall low-volatility of the ambient OA (Stark et al., 2017); the $C*=10^0$ $\mu$g m$^{-3}$ bin is assumed to have an initial concentration of 0 $\mu$g m$^{-3}$; Figure 2a shows an example of the initial ambient partitioning between the

volatility bins for a case from the BEACHON campaign. Monoterpene (MT) and isoprene concentrations are simulated explicitly outside of the VBS (though their reaction products enter the VBS as discussed earlier). Note that we do not include isoprene for the model runs from the BEACHON campaign due to the low contribution to measured SOA (1%) as compared to MT (20%, Palm et al., 2016). The isoprene concentrations (Karl et al., 2012; Kaser et al., 2013) were also consistently lower than the MT concentrations during BEACHON. Conversely, isoprene was observed to be the dominant measured

VOC during IOP2 of GoAmazon, with the average mass ratio of isoprene to MT during the dry season at 4.5 $\mu$g m$^{-3}$ per $\mu$g m$^{-3}$ (Palm et al., 2018), and thus isoprene is included in our model, even though isoprene's average contribution towards the predicted SOA during the dry season of GoAmazon was only 4% (Palm et al., 2018).

Data availability during BEACHON and GoAmazon caused data gaps that overlap some of the exposures modelled. For

these cases with missing measurement data, we assume concentrations; assumed values are listed in bold in Table 2. Each



assumed value is derived from either determining the trend from the nearest-available timepoints (for short data gaps) or by determining the concentration from different days with similar ambient conditions (for large data gaps).

### 2.3.3 Uncertain parameters

In order to understand the evolution of the size distributions of the OFR exposures from the BEACHON and GoAmazon field campaigns, we use TOMAS to explore the parameter spaces of five uncertain parameters. These parameters are: (1) the rate constant of gas-phase functionalization reactions with OH, (2) the rate constant of gas-phase ELVOC fragmentation reactions with OH, (3) the reactive uptake coefficient for heterogeneous fragmentation reactions with OH, (4) the nucleation rate constant for three different nucleation schemes, and (5) an effective accommodation coefficient that accounts for

possible particle diffusion limitations of aerosol particles larger than 60 nm in diameter. Table 3 lists each uncertain parameter, the assumed base value, and the parameter space that we search through for each parameter (the 'Multipliers' column).

As discussed in Sect. 2.3.1, we use as the base rate of $k_{OH}$ the relationship determined for aromatics by Jathar et al. (2014),

Eq. 1. As we are assuming that the products from the reactions of organic compounds in the VBS bins with OH drop by exactly one volatility bin per reaction (a 100-fold decrease in $C^*$) and there is uncertainty associated with the actual organic compounds (i.e. it is likely that the rates of reaction for some of the organic compounds are different than those of aromatics), we treat Eq. 1 as an uncertain parameter and we explore up to 10 times above and below this base equation. Jathar et al. (2014) determined the volatility-reactivity relationship of $k_{OH}$ for both aromatics and alkanes; our choice in using

the relationship for aromatics as a base case is arbitrary, as our parameter space encompasses both of the base values of $k_{OH}$ for aromatics and alkanes from their study.

In the model, we treat fragmentation reactions separately from the functionalization reactions. As discussed above, we select $1\times10^{-10}$ cm$^3$ molec$^{-1}$ s$^{-1}$ as the base value of the gas-phase fragmentation rate constant, $k_{ELVOC}$, and explore up to 9 times

above and below the base $k_{ELVOC}$. We note that this base fragmentation rate constant is one order of magnitude higher than the constant used in Palm et al. (2016) for BEACHON exposures. In their work, they used the rate constant for reactions with OH of an oxygenated molecule with no C=C bonds from Ziemann and Atkinson (2012) equal to $1\times10^{-11}$ cm$^3$ molec$^{-1}$ s$^{-1}$. They used this for their modelled LVOC concentration and assumed that 5 reactions of an LVOC with an OH molecule led to irreversible fragmentation into oxidized molecules that could no longer condense. Further, reaching $9\times10^{-10}$ cm$^3$

molec$^{-1}$ s$^{-1}$ for $k_{ELVOC}$ could exceed the kinetic limit for gas-phase fragmentation reactions. However, since we do not account for fragmentation reactions of higher-volatility species, a high $k_{ELVOC}$ value can be considered to effectively account for fragmentation reactions of higher-volatility species.



As previously discussed, for the reactive uptake coefficient $\gamma_{OH}$, we use a base value of 0.6, following the findings in Hu et al. (2016), and we explore up to 4 times above and below the base $\gamma_{OH}$ value, as previous studies have reported effective $\gamma_{OH}$ values ranging from $\leq 0.01$ to $> 1$ (Hu et al., 2016).

For our primary nucleation scheme, NUC1, (Eq. 4), we use a base nucleation rate constant value of $k_{NUC1}$ of $1\times10^{-21}$ cm$^6$

molec$^{-1}$ s$^{-1}$ and explore up to 20 times above and below the base $k_{NUC1}$ value. For our nucleation scheme sensitivity studies of NUC2 and ACT, (Table 3), we select base nucleation rate constant values of $1.25 \times10^{-14}$ cm$^3$ molec$^{-1}$ s$^{-1}$ and $2\times10^{-6}$ s$^{-1}$, respectively, and similarly explore up to 20 times above and below each base nucleation rate constant.

To account for possible particle-phase diffusion limitations, the effective accommodation coefficient is set to vary between 0.01 and 1 for particles larger than 60 nm in diameter (Table 3).

We simulate every combination of the uncertain parameters described above. In total, we run 10,125 sensitivity simulations for each BEACHON and GoAmazon OFR exposure for the first nucleation scheme (NUC1), going through each permutation for each of the five different uncertain parameters explored in this work. We further run 10,125 sensitivity simulations for both NUC2 and ACT for each experimental exposure. We acknowledge that there are further uncertainties in the measurements and modelling assumptions, including (1) potential but not modelled reactive uptake growth mechanisms,

(2) uncertainties in the reported OFR OH concentration, (3) isoprene chemistry that may affect NPF, (4) whether some products from gas-phase functionalization reactions decrease more or less in volatility per reaction than the assumed factor of 100 drop in volatility, and likely other factors. However, exploring these uncertainties is outside of the scope of this paper (and some of these are not entirely orthogonal to the uncertain factors explore here) and will be left to a future study.

**2.4 Description of cases**

**2.4.1 BEACHON-RoMBAS cases**

Figure 1 shows the measured initial and final SMPS volume size distributions for each exposure examined in this study from the BEACHON field campaign. We simulate these eight exposures between eq. ages 0.090 to 0.91 days in the TOMAS model for each combination of parameters (Table 3), initializing each run with the ambient conditions recorded at the time of each exposure (Table 2). Each modelled exposure was taken during the nighttime, when MTs were the dominant VOC. We

limit this study to exposures less than 1 eq. day of aging in order avoid the complications of modelling the different parameters in Sect. 2.3.3 across several orders of magnitude of OH, and since this is the range of exposures where NPF is most obvious experimentally.



### 2.4.2 GoAmazon2014/5 cases

In order to further test the validity of our results, we apply the TOMAS model version developed to simulate OFR exposures from the BEACHON field campaign to OFR exposures taken between August 31, 2014 and September 4, 2014 during the dry season of the GoAmazon field campaign. Figure S1 shows the initial and final SMPS volume size distributions for each

exposure examined in this study from the GoAmazon field campaign. We simulate each of these exposures for the same combination of parameters as used for the BEACHON simulations, initializing each run with the ambient conditions at the corresponding times (Table 2). However, unlike the BEACHON simulations, we include isoprene as a source of SOA in the model, with VBS yields given in Table 1. Again, like BEACHON, each modelled exposure was taken during the nighttime and are limited to exposures less than 1 eq. day of aging. During IOP2, it was observed that isoprene would peak during the

day around 3-4 pm local time and MT would peak later, around 6 pm local time (Liu et al., 2016; Martin et al., 2016). Isoprene was primarily depleted through oxidation reactions by nighttime but MT had a background level that remained approximately constant between midnight to noon (local times) when the concentrations would begin to rise again (Fig. S2). We model fewer exposures for GoAmazon than BEACHON (four vs. eight) as few of the GoAmazon OFR exposures during this time period showed significant SOA growth on top of the already-high ambient SOA concentrations as compared to

BEACHON. Also, many of the OFR exposures were either between 0.4-0.5 eq. days or >> 1 eq. day, so we were not able to cover as wide a range of <1 eq. day exposures as we did for BEACHON.

Bulk S/IVOCs were not measured during the GoAmazon campaign and instead we use the model to estimate the S/IVOC concentrations required to explain the aerosol particle growth. We use as base values of S/IVOC concentrations the average

MT:S/IVOC ratio from the BEACHON campaign, 1.4, as MT data is available during GoAmazon, and use the model to determine what S/IVOC concentrations are needed to help explain observed growth. This analysis is described in Sect. 3.2.

### 2.5 Description of simulation analyses

In order to determine the goodness-of-fit of each model simulation to the observed size distribution from the SMPS, we

compute the normalized mean error (NME) statistic of the first four moments of the size distribution for each model simulation:

$$NME = \frac{\sum_{i=0}^{4} \frac{|S_i - O_i|}{O_i}}{4}, \tag{5}$$

where $S_i$ and $O_i$ are simulated and observed $i$th moments. The $i$th moment is defined as

$$M_i = \int_0^\infty n_N D_p^i dD_p, \tag{6}$$



where $n_N$ is the number distribution and $D_p$ is the diameter range of the SMPS measurements, ~14-615 nm for the BEACHON campaign and ~14-710 nm for the GoAmazon campaign. The zeroth moment ($i = 0$) corresponds to the total number of particles, the first moment ($i = 1$) corresponds to the total diameter of particles (also referred to as the total aerosol

length), the second moment ($i = 2$) is proportional to the total surface area of particles, and the third moment ($i=3$) is proportional to the total volume of particles. Figure 3 gives an example of each measured final (OFR) moment (black solid line) as well as two different model runs' moments (colored lines) for a 0.23 eq. day aging exposure. The use of these four moments, including the less-common 1st "diameter" moment, allows us to include a broader range of the size distribution in the weighting rather than using just number or volume alone. An NME of 0 indicates a perfect fit between the simulation and

observations, an NME of 0.1 indicates that the average error of the four moments between the simulation and observations is 10%, and an NME of 1.0 indicates the average error of the four moments between the simulation and observations is 100%. Since the NME is taken as an absolute value, it does not give information on whether the model is on average overpredicting or underpredicting the moments; however, there could be model cases in which e.g., number and diameter are underestimated and surface area and volume are overestimated such that the NME statistic computed without the absolute

value (normalized mean bias, NMB) would be close to zero, falsely indicating a good fit despite the potentially large under- and overpredictions amongst the different moments. We determine each individual exposure's mean error of moments for both campaigns and further consider the average across all exposures for BEACHON and GoAmazon.

To determine the contribution to aerosol formation and growth for the OFR exposures studied here from the input VOCs vs

the input S/IVOCs, we compare the predicted change in the OFR in total aerosol particle number and volume between simulations with S/IVOCs to simulations with no S/IVOCs. We do this comparison for the six best-fitting simulations with S/IVOCs for each exposure and calculate the mean volume changes for these six simulations with and without S/IVOCs. With these number and volume changes, we calculate the fractional contribution of S/IVOCs to aerosol particle volume production in the OFR. We use the same technique to determine the contribution of isoprene to aerosol formation and growth

for the GoAmazon OFR exposures studied here using the same methods.

## 3. Results and Discussion

### 3.1 BEACHON-RoMBAS modelling results

#### 3.1.1 Average behaviour of exposures of eq. age 0.09 to 0.91 days for BEACHON-RoMBAS

Figure 4 represents the averaged NME summed across the eight 0.09-0.9 eq. day aging exposures modelled from the BEACHON field campaign, for the NUC1 $H_2SO_4$-organics nucleation scheme and the base value of the reactive uptake coefficient, $\gamma_{OH}$, of 0.6. (A discussion of the model sensitivity to other values of the reactive uptake coefficient is below.)




Figure 4 shows this average NME as a function of $a_{EFF}$ (effective accommodation coefficient of particles with diameters larger than 60 nm), $k_{ELVOC}$ (gas-phase ELVOC fragmentation rate constant), $k_{OH}$ (gas-phase functionalization rate constant), and $k_{NUC1}$ (rate constant for the first $H_2SO_4$-organics nucleation scheme). Lower $a_{EFF}$ values are necessary for the best fits; however, there are only slight differences between $a_{EFF} = 0.01$ and $a_{EFF} = 0.05$, and $a_{EFF} = 0.1$ (the left three columns,

respectively). Faster $k_{ELVOC}$ values are necessary for the best fits; however, similar to $a_{EFF}$, the base $k_{ELVOC}$ value (middle row), $3 \times k_{ELVOC}$, and $9 \times k_{ELVOC}$ values show similar results, with the regions of best fits shifting slightly with $k_{OH}$ and $k_{NUC1}$ values.

For the parameter combinations of $a_{EFF} = 0.01$ through $a_{EFF} = 0.05$ and $9 \times k_{ELVOC}$ (the top row of Figure 4), the $2 \times k_{NUC1}$ and

$4 \times k_{NUC1}$ values have the best fits. These $2 \times k_{NUC1}$ and $4 \times k_{NUC1}$ values are similar to those found by Riccobono et al. (2014) for experimental conditions at 278 K (a $k_{NUC1}$ value of $3.27 \times 10^{-21}$ $cm^6$ $molec^{-1}$ $s^{-1}$). However, the other wells of good fits for the base $k_{ELVOC}$ value and $3 \times k_{ELVOC}$ have lower nucleation rate constants than that of Riccobono et al. (2014). As mentioned earlier, these $k_{NUC1}$ values determined here correspond to the temperatures of the measurements (between 282-290 K; Table 2) which is 4-12 K warmer than the experimental conditions of Riccobono et al. (2014), hence we may expect lower $k_{NUC1}$

values due to the temperature dependence of nucleation (Yu et al., 2017). Figure 4 shows that the wells of best fits for all parameter combinations require slightly higher $k_{OH}$ values than the base $k_{OH}$ (based on the $k_{OH}$ values from Jathar et al. (2014)), usually on the order of 1.5-2.5 times higher.

Figures 2b and 3 show an example of the final volatility distribution and size distributions for the best-fit case for an

exposure of 0.23 eq. days, corresponding to the model parameters of $2 \times k_{NUC1}$, $5 \times k_{OH}$, $0.5 \times y_{OH}$, $k_{ELVOC}$, and $a_{EFF} = 0.01$. Figure 2a and b gives the initial and final partitioning for this case, respectively, showing that virtually all of the initial gas-phase S/IVOCs have reacted with OH to either enter the lower volatility bins or to fragment into VOC products no longer tracked in the model. Figure 3 shows each modelled moment compared to each observed moment of the size distribution used in calculating the NME for the best-fit case.

Figures S3, S5, S7, S9, S11, S13, S15, and S17 show the same analysis as presented in Figure 4 for each individual exposure modelled for the base value of $y_{OH}$, 0.6. Figures S4, S6, S8, S10, S12, S14, S16, and S18 plot each observed final (OFR output) moment used in computing the NME statistic (number, diameter, surface area, and volume) compared to the six TOMAS cases with the lowest (best) NME statistic and six TOMAS cases with the highest (worst) NME statistic. For

comparison, the observed initial (ambient air) moments are also plotted for each moment.

Figure S19 shows the same analysis as Fig. 4, but for the NUC2 nucleation scheme. It is qualitatively quite similar to NUC1 but with the wells of averaged best-fit regions shifted and expanded slightly for some cases. Since we do not have measurements to further constrain the system, we acknowledge that we cannot definitively select NUC1 or NUC2 as being





the better nucleation parameterization and instead note that both nucleation schemes appear to provide physically-meaningful results and require further study. In contrast, Fig. 5 shows the same analyses of Fig. 4 but for the ACT nucleation scheme (Eq. 5). Figure 5 shows that there are regions of moderate NME values between 0.45-0.5 for $a_{EFF}$ = 0.01 through $a_{EFF}$ = 0.05. These regions of moderate fits occur for higher values of $A$ (between 4 and 20×$A$) for a wide range of $k_{OH}$

values. The best fits are seen for higher values of $k_{ELVOC}$ (between the base value of $k_{ELVOC}$ and 9×$k_{ELVOC}$), the highest nucleation rates (for values of $A$ between 10-20×$A$) and lower to mid rates of $k_{OH}$ (in general between 0.4×$k_{OH}$ and the base value of $k_{OH}$). In general, we do not see as good of fits as we do for the NUC1 and NUC2 schemes; however, it does appear that for some combinations of parameters, a reasonable model-to-measurement fit can be achieved with an activation nucleation scheme. Thus, we conclude that for this study, the $H_2SO_4$-organics mediated nucleation schemes fit the

measurements better than the activation nucleation scheme in our model for the OFR measurements taken during the BEACHON campaign.

Further, as the best fits in the model come from the $H_2SO_4$-organics mediated nucleation schemes, and the best-fit $k_{NUC}$ values are similar to those of Riccobono et al. (2014) where particle-phase chemistry was likely unimportant (low aerosol

volume), this is indicative evidence that the creation of gas-phase ELVOCs through oxidation reactions could be dominant over the creation of particle-phase ELVOCs (either through accretion reactions and/or acid-base reactions) for the OFR present at BEACHON campaign, as high concentrations of gas-phase ELVOCs are necessary to facilitate nucleation. It is however important to note that we are limited in our confidence of the actual values of the best fits of the different nucleation rate constants ($k_{NUC1}$, $k_{NUC2}$, and $A$), since each nucleation scheme is sensitive to the concentration of sulfuric acid, and in the

majority of the exposures modelled we did not have a direct measurement of $SO_2$ available for all cases and instead had to estimate $SO_2$ concentrations for nearly half of the cases.

It is of note that in general, the simulations using $a_{EFF}$ = 0.5 and $a_{EFF}$ = 1.0 do not yield good fits for any of the nucleation schemes tested here, indicating the importance of some sort of process that limits uptake to the larger aerosol. Figure 3

illustrates the impact of the effective accommodation coefficient for a 0.23 eq. day aging exposure: it shows each of the first four moments of the size distribution for the initial and final observations (dotted and black lines) and for the best-fit case for this exposure (solid blue lines) and the model simulation with the same best-fit parameter values but for $a_{EFF}$ = 1.0 (dashed orange lines). Compared to the final observations, the best-fit case closely matches the changes in each moment for the Aitken and accumulation modes. However, the best-fit case with $a_{EFF}$ set to 1.0 clearly overestimates growth for the

accumulation mode and underestimates growth for the Aitken mode. In general, when $a_{EFF}$ = 1.0 there was no combination of the other parameters tested that could simultaneously capture (1) the number and growth of the growing nucleation mode and (2) the change in volume of the large mode. When $a_{EFF}$ = 1.0, either the new particles did not grow enough or the large particles grew too much throughout our parameter space. Hence, we were unable to explain the observations without limiting the uptake of material to particles with diameters larger than 60 nm. Additionally, when we tried to lower the





accommodation coefficient of smaller particles (not shown), we could not simulate the growth of these particles. While our scheme for limiting the uptake of vapors to the large particles is very simple in this study, we feel that some limitations of vapor uptake to accumulation-mode particles must be at play, possibly from particle-phase diffusion limitations or other reasons. Zaveri et al. (2017) modelled the controlled bimodal growth of aerosol from isoprene and α-pinene oxidation

products and found that in order to replicate the observed growth, both the Aitken and accumulation modes required particle-phase diffusivity limitations. However, their experimental conditions were at much lower humidity than the BEACHON exposures, and did not include any other atmospheric species that could be relevant at BEACHON.

The BEACHON simulations show very little sensitivity towards the reactive uptake coefficient ($\gamma_{OH}$) parameter, regardless of which nucleation scheme was used. Figure S20 shows the model sensitivity towards $\gamma_{OH}$: the figure is for the NUC1 nucleation scheme and base value of $k_{ELVOC}$. Across each row, the effective accommodation increases and down each column, $\gamma_{OH}$ increases. Within each subplot, the rate constant of gas-phase reactions with OH increases along the x-axis and the rate constant for nucleation increases along the y-axis. Isolating $\gamma_{OH}$ (each column) shows that for a given set of the other

four parameters, the varying values of $\gamma_{OH}$ do not significantly change the NME. Thus, it would appear that gas-phase fragmentation reactions dominate over particle-phase fragmentation reactions in the OFR for exposures less than one day of equivalent aging. This is in agreement with previous studies of heterogeneous mass loss in OFRs; Hu et al. (2016) did not see significant loss of aerosol mass until exposures greater than 1 day eq. aging for OFR data collected during both the Southern Oxidant and Aerosol Study (SOAS) and the GoAmazon campaign. Because of this, we will focus the remaining

discussion upon runs using only the base value of $\gamma_{OH}$, 0.6.

### 3.1.2 Importance of S/IVOCs for SOA formation at BEACHON-RoMBAS

Palm et al. (2016) compared the total SOA formed in the OFR during the BEACHON campaign to the predicted yield from the measured VOCs for OH oxidation in the OFR. For the analysis, they included the measured MT, sesquiterpene (SQT),

isoprene, and toluene concentrations and used low-NO$_x$ (to match the OFR chemical regime, Li et al., 2015; Peng et al., 2015), OA-concentration-dependent chamber-derived particle yields for each species. They determined that MTs contributed on average 87% of the SOA predicted to form from these VOC precursors, but on average, the maximum measured SOA formation was 4.4 times higher than the predicted SOA formation. Palm et al. (2016) attributed the yield from measured S/IVOC concentration to the mass difference between measured and predicted SOA yields and concluded that OH oxidation

of organic gases could potentially produce approximately 3.4 times more SOA from S/IVOC gases than from the measured VOCs, by using SOA yields for S/IVOC that were consistent with the literature. The correlation between measured SOA formation and ambient MT concentrations was $R^2$=0.56, indicating that the S/IVOCs controlling SOA formation in the OFR were primarily related to MT and other biogenic gases with similar diurnal behavior.



To determine the contribution towards the change in total number and volume, we compare the changes in total volume between the averaged change in total volume for the six cases with the lowest (best) NME values of the original model runs for the NUC1 nucleation scheme to the same six cases (matching parameters) but with the initial S/IVOC concentration set to zero (See Sec. 2.5 for calculation details). Table 4 summarizes the fractional contribution of the measured initial S/IVOCs (Table 2) towards the total change in number and volume. The model predicts that the S/IVOCs contribute on average 85% towards the total new number formed in the OFR, indicating a strong dependence on S/IVOCs for new particle formation in the OFR at BEACHON. The contribution of S/IVOCs towards the total change in volume is lowest for the lowest exposures, and increases with increasing eq. age of each exposure. This is primarily due to the increasing equivalent timescales of the increasing OH exposures: within our model it takes more reactions with OH for S/IVOC species to reach the lowest volatility bins than the MT and isoprene species. Thus with increasing timescales (or eq. ages), the contribution of S/IVOCs towards SOA formation and growth will increase as a higher fraction of these species reach the lowest volatility bins; the results in Table 4 corroborate this. However, given that the chemical evolution of S/IVOC is probably more complex than is represented here, we do not know if this result of S/IVOCs contributing a lower fraction of volume for low exposures is a robust conclusion. Overall, we predict that the average fractional contribution of the initial ambient S/IVOCs towards the change in total volume is 39% for the BEACHON exposures, and that the initial ambient MT contributes the remaining 61% towards the change in total volume. Palm et al. (2016) and Hunter et al. (2017) estimated from two independent analyses that S/IVOCs contributed on average 77-78% towards the total mass SOA formation during BEACHON. It is likely that part of the difference between our model findings and Palm et al. (2016)'s findings is due to the difference in number of samples examined between the two studies as well as differences in the length of exposures analyzed, since Palm et al. (2016) included multi-day exposures in their analysis. It is important to note that running the model with the initial S/IVOCs set to zero ("S/IVOCs off") does not perfectly inform us of the theoretical SOA yield of the MT concentration because the overall particle-phase yield of MTs products decreases with S/IVOCs off due to less mass to partition to.

### 3.2 GoAmazon2014/5 modelling results

In order to model GoAmazon size distributions with TOMAS, we assumed an initial S/IVOC concentration, as no instrumentation was present during the campaign to measure total S/IVOC mass. For a starting total S/IVOC concentration, we used the same measured ratio of S/IVOCs to MTs from BEACHON of 1.4 (Table 2). This initial S/IVOC concentration was not sufficient to explain the observed change in aerosol volume, nucleation, and new-particle growth in the OFR for GoAmazon (see Figs. S21-S22 for an example). We found that the initial S/IVOC concentration needed to be increased by between 20-40 times in order to fit the observed distributions. As BEACHON was dominated by biogenic emissions (primarily MTs), but GoAmazon had major contributions from anthropogenic and biomass burning sources as well as various biogenic emissions (Palm et al., 2018), the larger S/IVOC is thought to be dominated by emissions and partially





oxidized products from the two latter sources. We present results for 30 times the base S/IVOC concentrations (Table 2) in Figs. 6 and S23-S32 as this amount of increase showed consistently good results across the four exposures modelled. For comparison, the total initial S/IVOC mass for the BEACHON OFR exposures modelled ranges between 2.89 and 14.02 $\mu$g m$^{-3}$ whereas the total initial S/IVOC mass for the GoAmazon OFR exposures modelled ranges between 9.0 and 18.6 $\mu$g m$^{-3}$

when the assumption of 30 times higher S/IVOC:MT ratios is used. Hence, even though the S/IVOC:MT ratios were higher for GoAmazon relative to BEACHON, our assumed S/IVOC concentrations were in the same general range for the two campaigns. We note that by not including the measured concentrations of SQT, benzene, toluene, xylenes, and trimethylbenzenes in our model likely slightly biases our S/IVOC estimation high, but not by a significant amount as Palm et al. (2018) found that these species contributed on average a sum total of 8% towards the measured SOA yield from the

measured VOC precursor species.

Figure 6 represents the averaged NME across the four 0.3-0.6 eq. day aging exposures modelled from the GoAmazon field campaign for the NUC1 H$_2$SO$_4$-organics nucleation scheme and the base value of $\gamma_{OH}$, 0.6. In general, there are wider ranges of $k_{OH}$, $k_{ELVOC}$, and $k_{NUC1}$ values that give small NMEs for the averaged GoAmazon modelled exposures than for the averaged

BEACHON modelled exposures. The model simulations generally perform best with lower accommodation coefficients of the larger particles (between $a_{EFF} = 0.01$ and $a_{EFF} = 0.1$), similar to the BEACHON results; however, there are some similarly low-NME results between $a_{EFF} = 0.05$ and $a_{EFF} = 1$ for the two highest $k_{ELVOC}$ values. Bateman et al. (2015) showed that submicrometer PM aerosol in the Amazon rainforest measured at the same T3 site as the GoAmazon campaign during the dry season tend to be liquid, so it is possible that the uptake/diffusion limitations to the accumulation mode inferred for

BEACHON may not occur during GoAmazon. However, we do not have enough information to learn more about the causes of uptake/diffusion limitations to the accumulation mode or differences between the campaigns.

Previous ambient observations of the Amazon rainforests have not observed nucleation at the surface (e.g., Spracklen et al., 2006; Martin et al, 2010; Kanawade et al., 2011). Reasons could include low sulfuric acid (Kanawade et al., 2011), high

condensation sinks resulting from a strong source of primary biogenic aerosols during the dry season (Lee et al., 2016), and possible yet currently unexplained suppression mechanisms from isoprene and its oxidation products (Lee et al., 2016), the dominant biogenic VOC of the region (Guenther et al., 2012). Wang et al. (2016) found high concentrations of small particles in the lower free troposphere during the wet season of GoAmazon; however, they found that these particles appeared to be from NPF and subsequent condensational and coagulational growth from the outflow regions of deep

convective systems, such as those common to the Amazonian rainforest during the wet season. These particles could then be transported to the boundary layer through vertical transport. By contrast, in some of the OFR-oxidized air during the GoAmazon campaign the size distributions show clear evidence of NPF and growth (e.g., Fig. S1) and the TOMAS model simulations corroborate the observed NPF (Figs. S26, S28, S30, S32), even at the initial S/IVOC inputs (Fig. S22). The OFR





shifts the relative timescales of chemistry versus condensation, which may create higher concentrations of low-volatility vapors capable of participating in nucleation and early growth relative to the ambient atmosphere during GoAmazon. The lowest NME values (best fits) from the averaged BEACHON modelled exposures (Fig. 4) for the highest two $k_{ELVOC}$ values overlap regions of wells of best fits for the averaged GoAmazon modelled exposures. For GoAmazon there is a wider range

of $k_{OH}$, $k_{ELVOC}$, and $k_{NUC1}$ values that give low NME values compared to BEACHON modelled exposures. We note that the lower number of exposures modelled for GoAmazon than modelled for BEACHON limit our confidence in comparing the two campaigns' results to each other, as does the narrower range of equivalent aging (between 0.39 and 0.52 eq. days aging for GoAmazon compared to 0.09 to 0.91 eq. days aging for BEACHON). Figures S25, S27, S29, and S31 show the same analysis as presented in Fig. 6 for each individual exposure modelled for the base value of $\gamma_{OH}$, 0.6. Figures S26, S28, S30,

and S32 plot each observed final size distributions for the first four moments (solid black lines) used in computing the NME statistic compared to the six TOMAS cases with the lowest (best) NME statistic and six TOMAS cases with the highest (worst) NME statistic. For comparison, the observed initial (ambient) moments are also plotted for each moment.

Tests of NUC2 and ACT show similar changes from NUC1 for GoAmazon as BEACHON (Figs. S23 and S24). NUC2

results were qualitatively similar to NUC1, and we cannot determine which scheme performed better. The regions of lowest NME values (best fits) shifted for the ACT scheme relative to the NUC1 and NUC2 schemes, and generally the NMEs are not quite as low as for NUC1 and NUC2, although better fits are found for the ACT nucleation scheme for GoAmazon than BEACHON. Thus it would seem that either a $H_2SO_4$-organics mediated nucleation scheme or a $H_2SO_4$-only nucleation scheme can be used in our model to describe the OFR measurements taken during the GoAmazon campaign. Like

BEACHON, we are still limited in our confidence of the actual values of the best fits of the different nucleation rate constants ($k_{NUC1}$, $k_{NUC2}$, and $A$) as each nucleation scheme is sensitive to the concentration of sulfuric acid, and some exposures had an estimated $SO_2$ concentration.

Similar to BEACHON, more good fits for each nucleation scheme occur at lower values of $\alpha_{EFF}$, again pointing to the

potential importance of vapor-uptake/diffusion limitations at least within the OFR timescales. Again, varying the reactive uptake coefficient was not seen to significantly change the NME values of each set of parameter values, regardless of nucleation scheme, and thus we only show results for the base value of the reactive uptake coefficient (Figs. 6, S23-S24, S25, S27, S29, and S31).

**3.2.1 Importance of S/IVOCs for SOA formation at GoAmazon2014/5**

Unlike the BEACHON campaign, bulk S/IVOCs were not measured directly during the GoAmazon campaign. However, Palm et al. (2018) applied a similar analysis to that of Palm et al. (2016) to determine the measured vs. predicted SOA yield. They found that on average, OH oxidation of ambient air during the GoAmazon campaign (dry season) produced 6.5 times





more SOA than could be accounted for from the measured ambient VOCs. They used the low-NO$_x$ SOA yields (verified by standard addition during the campaign) corresponding to the expected conditions in the OFR (Li et al., 2015) for the measured MT, SQT, toluene, and isoprene concentrations. Unlike BEACHON, it was observed for GoAmazon that the slope of the measured versus predicted SOA formation from OH oxidation varied as a function of time of day, with predicted SOA

lower during the nighttime than daytime but measured SOA formation higher during the nighttime than daytime. Palm et al. (2018) were uncertain of the reasons for the observed SOA trends, but hypothesize that several processes likely play a role, including diurnal changes in emissions, boundary layer dynamics, and variable ambient oxidant concentrations. Palm et al. (2018) hypothesized that, like the BEACHON campaign, S/IVOCs could make up the mass difference between measured and predicted SOA yields from OH oxidation in the OFR. In this study, it was found that between 20-40 times more initial

S/IVOCs than the base concentrations of S/IVOCs (Table 2) derived from using the ratio of S/IVOCs:MT, 1.4, from BEACHON was required to explain the aerosol formation and growth and change in total volume observed in the OFR during GoAmazon for OH oxidation. This corroborates the findings of Palm et al. (2018) that no strong correlation was found between any one VOC precursor gas, indicating that SOA formation was impacted by multiple sources.

To determine the contribution of MT and isoprene towards the change in total number and volume for the GoAmazon exposures, we repeat the analysis done for the BEACHON exposures (Sect. 3.1.2.) and the results are summarized in Table 4, using the S/IVOC concentrations of 30 times the base S/IVOC concentrations. The model predicts that the optimized S/IVOC concentrations contribute 100% towards the new aerosol number formation observed for each exposure modelled, again pointing towards the importance of S/IVOCs for NPF in the OFR. However, since SQT, benzene, toluene, xylenes, and

trimethylbenzenes (all measured ambient VOC species predicted to contribute towards SOA formation) were not included in the model, we cannot conclude that S/IVOCs are actually responsible for 100% of the new aerosol formed in the OFR. Similar to BEACHON, the fractional contribution of S/IVOCs towards the change in total volume increases with increasing eq. age; overall, the average fractional contribution of the best-fit S/IVOC concentration towards the change in total volume is 0.66. By comparison, Palm et al. (2018) found that the fractional contribution of S/IVOCs towards the measured SOA

formation during the dry season of GoAmazon was on average 0.85. We again expect that the VOCs will have artificially low SOA yields in the "S/IVOCs off" simulations, indicating that MT and/or isoprene could contribute more towards the change in total volume than indicated here. However, Palm et al. (2018) found that the yield dependence of ambient SOA precursors on ambient OA was weaker for the OFR GoAmazon exposures than it was for the chamber-derived parameterizations used to predict the OFR yield.

### 3.2.2 Importance of isoprene for SOA formation at GoAmazon2014/5

The TOMAS box model does not include isoprene-specific oxidation pathways and instead allows it to oxidize in the VBS scheme along with the other lumped oxidized species. We determine the fractional contribution of the initial isoprene





concentration towards the change in total number volume for each exposure modelled (Table 5); the remaining fraction is the total volume change attributable from initial MT and the optimized initial S/IVOC concentrations (30 times that of the base S/IVOC concentrations). At maximum, it is predicted that within the OFR isoprene will contribute 0% and 3% towards the change in total number and volume, respectively; on average, it is predicted that isoprene will contribute 0% and 1% towards

the change in total number and volume. However, this does not preclude the potential importance of isoprene towards ambient SOA formation. The OFR can only form SOA from the gases that enter it; although isoprene emissions are high, isoprene reacts quickly (Atkinson and Arey, 2003b) so that much of the potential SOA from isoprene and its oxidation products enters the chamber already in the particle phase. Further, the OFR does not capture the most important isoprene SOA formation pathway, such as IEPOX-SOA produced from reactive uptake on time scales longer than the OFR residence

time (Hu et al., 2016). Palm et al. (2018) estimated that on average during the dry season, isoprene contributes 5% towards the predicted SOA mass yield from the measured ambient VOC precursor species in the OFR. Not including isoprene-specific oxidation pathways in our model may be a source of error in calculating the contribution of isoprene towards the total change in number and mass.

**4 Conclusions**

In this study, aerosol size distributions between 0.09-0.9 days of eq. aging formed under OH oxidation in an OFR during the BEACHON-RoMBAS (BEACHON) and GoAmazon2014/5 (GoAmazon) field campaigns were modelled in the TOMAS box model in order to better-understand the microphysical processes that shape the size distribution under oxidative aging. We explored the following parameter spaces to find regions of best-fit model-to-measurement agreements: (1) nucleation

rate constants for two $H_2SO_4$-organics nucleation mechanisms versus a $H_2SO_4$ activation nucleation mechanism, gas-phase (2) functionalization and (3) fragmentation rate constants, (4) heterogeneous reactions with OH resulting in fragmentation and aerosol mass loss, and (5) potential particle diffusion limitations to the accumulation mode.

In order to limit the scope of this study, several uncertain processes and values were not included in this analysis. We did not

include the formation of low-volatility organics through particle-phase acid-base reactions or accretion reactions, as (1) no measurements of gas-phase bases were made at either campaign and (2) the model results indicate the importance of the gas-phase ELVOC creation pathway must be fast in order to drive nucleation, which may limit the importance of particle-phase pathways. We did not consider the model sensitivity towards the input OH concentration, although there is uncertainty associated with the estimated OH exposure (Palm et al., 2016; 2018). We further did not explore the model sensitivity

towards the assumed decreased of a factor of 100 in volatility for each product from OH functionalization reactions, nor did we explore the sensitivity of including fragmentation reactions for volatility bins higher than the ELVOC bin. These two uncertainties are not entirely orthogonal to the uncertainties in $k_{OH}$ and $k_{ELVOC}$ that we did explore, and including them would have increased the number of free parameters in the model, making it more challenging to determine what combinations of





parameters most-closely match the actual processes occurring in the OFR. Finally, there is evidence for possible NPF suppression in some isoprene-dominated regions but as those mechanisms are as yet unknown (Lee et al., 2016), no isoprene chemistry was explicitly simulated for the modelled GoAmazon exposures. However, as shown in Table 5, isoprene was only a minor contributor to our predicted aerosol volume for the GoAmazon simulations.

We found that we could not explain the observed size-distribution shift without slowing the uptake of SOA to the accumulation-mode particles. With an accommodation coefficient of 1 assumed for the full size distribution, these larger particles underwent too much condensational growth relative to the nucleation mode for all test cases. We speculate that this slowed uptake of larger particles may be indicative of particle-phase diffusion limitations. We approximate vapor-uptake

limitations by allowing the accommodation coefficient of particles larger than 60 nm diameter to vary between 0 and 1. We found that we can achieve the best fits of the size distribution when the accommodation coefficient of these larger particles was 0.1 or lower (if we similarly lowered the accommodation coefficient of smaller particles, we would not have gotten good fits as the new particles did not grow enough). Whether this is representative of ambient aerosol processes or just representative of conditions within the OFR is the subject of a future study.

We found that gas-phase fragmentation reactions also had a significant impact upon the modelled size distributions. Our best-fit gas-phase fragmentation rate constants were higher than that of a previous mass-based study of OFR exposures from BEACHON (Palm et al., 2016) required to model the distributions. However, these higher rates may be because our model only simulated fragmentation reactions of the lowest volatility compounds, that of $C^* \leq 10^{-4}$ $\mu$g m$^{-3}$, whereas in reality

fragmentation reactions can occur to higher-volatility compounds (although the likelihood of fragmentation likely increases with decreasing volatility). Thus, the higher fragmentation rate constant can be seen as compensating for fragmentations of higher-volatility compounds. Including fragmentation of higher volatility species would lower the fraction of the organic vapors that then make it to lower volatility. This would then potentially decrease nucleation rates and slow the growth rates of the smallest particles. However, the inclusion of a more-complex fragmentation scheme would have added more free

parameters to our study and will be left to a future study.

In general, the H$_2$SO$_4$-organics nucleation mechanisms performed better than the activation nucleation mechanism for both campaigns. We found that the nucleation rate constants for the H$_2$SO$_4$-organics nucleation mechanism suggested by Riccobono et al. (2014) allowed for good models fits, with the caveat that the temperatures of both campaigns were higher

than the experimental conditions of Riccobono et al. (2014) (4-12 K higher for BEACHON and 18-19 K higher for GoAmazon). Similarly, we found that gas-phase oxidation rate constants similar to that of Jathar et al. (2014), fit from aromatics, allowed for good fits (we assumed that these reactions were 100% functionalization and treated the fragmentation reactions separately). Finally, we found that heterogeneous reactions of the OA with OH resulting in fragmentation and





aerosol mass loss did not appear to significantly impact the distributions modelled in this study. As all of our equivalent exposure times tested were less than one day, these results are consistent with previous OFR studies on heterogeneous aging that found that heterogeneous losses of OA from OH were not important for these exposure timescales (Hu et al., 2016). Like Palm et al. (2016; 2018), our results indicate the importance of S/IVOCs towards aerosol growth in the OFR at both the

BEACHON and GoAmazon campaigns. We find that S/IVOCs contribute on average 85% and 39% (BEACHON) and 100% and 66% (GoAmazon) towards the change in total number and volume, respectively, for the exposures modelled in this study.

This study has shown the potential for using OFRs to study factors that control NPF and size-distribution evolution using

ambient-air mixtures. The fact that coagulation plays a small role in the measured number concentration indicates that this type of reactor is useful to evaluate model parameterizations of the number of nucleated particles and their growth, as a function of ambient and OFR conditions. Using an OFR greatly expands the parameter space over which comparisons can be made as well as the number of cases that can be studied, compared to using only ambient data where parameter variations are more narrow, and where NPF is not observed under many conditions. Future studies could use OFRs in nucleation studies to

both better-understand the dependencies of nucleation on input species (e.g., $H_2SO_4$, gas-phase bases, and specific VOCs) by injecting controlled amounts of each species or precursors on top of ambient air at variable oxidant concentrations, as well as determine dominant nucleation mechanisms for different ambient environments. In order to assist in ambient nucleation studies, more-precise measurements of ambient $SO_2$ should be made during ambient campaigns in order to more accurately test current nucleation theories (all of which depend upon the concentration of $H_2SO_4$) against different ambient

environments. Measurements of $H_2SO_4$ and ELVOCs, as well as bases such as ammonia and amine species, inside the reactor would help constrain the nucleation and growth mechanisms significantly. Additionally, studies focused on size-distribution evolution processes could include size-dependent particle-phase composition and property measurements in order to assess parameters such as particle phase state and presence of acid-base or accretion products as a function of equivalent aging in order to better constrain the model assumptions against observations. Focusing on lower OH exposures

(<<1 day, to limit fragmentation reactions prior to condensation/nucleation) as well as varying OFR residence times may allow extracting more information on new-particle formation and growth from these experiments. Another vein of research could use the best-fit parameters found in this study and similar studies to initialize ambient models in order to predict under what conditions (emissions, initial particle concentrations, OH concentrations, and so forth) one would anticipate NPF and growth. Such predictions, if well-validated by corresponding ambient measurements, could help construct simple

parameterizations for use in regional and global models to better-simulate NPF and growth events in order to improve predictions of size-resolved aerosol concentrations and their corresponding impacts upon climate and health.




**Data availability**

The data used from the BEACHON-RoMBAS campaign in the publication are available at http://manitou.acom.ucar.edu/ (https://doi.org/10.5065/D61V5CDP ). The data sets used from the GoAmazon2014/5 campaign in this publication are available at the ARM Climate Research Facility database for the GoAmazon2014/5

campaign (https://www.arm.gov/research/campaigns/amf2014goamazon). Most data shown in the figures in this paper (including Supplement) pertaining to measurement results can be downloaded from http://cires1.colorado.edu/jimenez/group_pubs.html. All data shown in the figures pertaining to model results in this paper (including Supplement) are available upon request and will be posted on a web repository upon final publication of the paper.

**Competing interests**

The authors declare they have no conflict of interest.

**Acknowledgements**

This research was supported by the US Department of Energy's Atmospheric System Research, an Office of Science, Office of Biological and Environmental Research program, under Grant No. DE-SC0011780, and by the U.S National Oceanic and Atmospheric Administration, an Office of Science, Office of Atmospheric Chemistry, Carbon Cycle, and Climate Program, under the cooperative agreement award #NA17OAR430001. The CU-Boulder group was supported by US DOE (BER/ASR)

DE-SC0016559, and US EPA STAR 83587701-0. This manuscript has not been formally reviewed by EPA. The views expressed in this document are solely those of the authors and do not necessarily reflect those of the Agency. EPA does not endorse any products or commercial services mentioned in this publication. Institutional support was provided by the Central Office of the Large Scale Biosphere Atmosphere Experiment in Amazonia (LBA), the National Institute of Amazonian Research (INPA), and Amazonas State University (UEA). We acknowledge support from the Atmospheric Radiation

Measurement (ARM) Climate Research Facility, a user facility of the United States Department of Energy (DOE), Office of Science, sponsored by the Office of Biological and Environmental Research, and support from the Atmospheric System Research (ASR) program of that office. Additional funding was provided by the Amazonas State Research Foundation (FAPEAM), the São Paulo State Research Foundation (FAPESP), the USA National Science Foundation (NSF), and the Brazilian Scientific Mobility Program (CsF/CAPES). The TD-EIMS measurements were supported by NOAA grant

NA10OAR4310106 (MIT). The research was conducted under scientific license 001030/2012-4 of the Brazilian National Council for Scientific and Technological Development (CNPq). PTR-TOF-MS measurements were supported by the Austrian Science Fund (FWF) project no. L518-N20, L. Kaser received a DOC-fForte-fellowship of the Austrian Academy of Science. We are grateful to Andrew Turnipseed for $SO_2$ measurements, to Lisa Kaser and Steven Sjostedt for PTR-TOF-MS measurements, and to Shantanu Jathar and Scot Martin for useful discussions.





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





**Tables**

**Table 1:** Product fractional mass yields for lumped monoterpenes and isoprene (GoAmazon only) in each VBS bin in TOMAS. The monoterpene yields are based on Henry et al. (2012), with the yield for the $C*=10^{-4}$ bin representing the

5   average yield from oxidation of OH of the monoterpene species examined in Jokinen et al. (2015). The isoprene yields are from Tsimpidi et al. (2010), remapped to fit the TOMAS model's bin scheme, with the yield to the $C*=10^{-4}$ bin from isoprene OH oxidation from Jokinen et al. (2015).

| Species | Aerosol yield per bin $[\log(C*)]$ | | | | | |
|---|---|---|---|---|---|---|
| | -4 | -2 | 0 | 2 | 4 | 6 |
| Monoterpene | 0.0075 | 0.00005 | 0.083 | 1.095 | 0.125 | 0.0 |
| Isoprene | 0.0003 | 0.0 | 0.023 | 0.03 | 0.0 | 0.0 |





**Table 2:** All BEACHON-RoMBAS inputs (values where measurements are missing and we estimated values are in bold) and GoAmazon2014/5 inputs (assumed values in bold). Each value represents the ambient condition present at the beginning of each modelled exposure. The OH conentration is calculated by assuming that 1 day of aging is equal to a 24 hour average atmospheric OH concentration of $1.5 \times 10^6$ molec day $cm^{-3}$ and that the average residence time of the OFR was 134 s at BEACHON-RoMBAS and 171 s at GoAmazon2014/5.

| Exposure in eq. age, days (OH conc., cm$^{-3}$) | MT [µg m$^{-3}$] | Isoprene [µg m$^{-3}$] | SO$_2$ [ppb] | S/IVOC [µg m$^{-3}$] | Total Mass [µg m$^{-3}$] | OA/total mass ratio | Temperature [K] | RH [%] |
|---|---|---|---|---|---|---|---|---|
| BEACHON-RoMBAS | | | | | | | | |
| 0.090 ($8.7 \times 10^7$) | **9.09** | n/a | 0.02 | 8.09 | 3.22 | 0.85 | 284 | 92 |
| 0.098 ($9.5 \times 10^7$) | 8.97 | n/a | 0.029 | 2.89 | 2.47 | 0.8 | 282 | 82 |
| 0.16 ($1.5 \times 10^8$) | 8.94 | n/a | **0.029** | **10** | 1.52 | 0.79 | 290 | 73 |
| 0.23 ($2.2 \times 10^8$) | **9.09** | n/a | **0.029** | 9.3 | 3.4 | 0.84 | 288 | 91 |
| 0.27 ($2.6 \times 10^8$) | 9.09 | n/a | 0.029 | **10** | 1.6 | 0.79 | 289 | 84 |
| 0.77 ($7.4 \times 10^8$) | **3.6** | n/a | **0.029** | 6.9 | 2.24 | 0.9 | 286 | 94 |
| 0.82 ($7.9 \times 10^8$) | **9.09** | n/a | **0.079** | 14.02 | 3.17 | 0.85 | 286 | 91 |
| 0.91 ($8.8 \times 10^8$) | **9.09** | n/a | 0.029 | 10.85 | 3.66 | 0.86 | 287 | 92 |
| GoAmazon2014/5 | | | | | | | | |
| 0.39 ($2.6 \times 10^8$) | 0.56 | 0.86 | 0.14 | 0.40[a] | 4.85 | 0.88 | 296 | 102 |
| 0.40 ($3 \times 10^8$) | 0.42 | 0.90 | 0.06 | 0.30[a] | 4.94 | 0.88 | 296 | 101 |
| 0.51 | 0.68 | 1.34 | **0.11** | 0.49[a] | 8.7 | 0.81 | 297 | 99 |





| | | | | | | | | |
|---|---|---|---|---|---|---|---|---|
| $(3.9\times10^8)$ | | | | | | | | |
| 0.53 $(4\times10^8)$ | 0.87 | 1.17 | **0.11** | 0.62[a] | 8.17 | 0.8 | 297 | 99 |

[a]S/IVOCs were not measured during GoAmazon2014/5. The average BEACHON-RoMBAS campaign MT:S/IVOC ratio was 1.4; this ratio was used to create an initial S/IVOC amount. See text for more details.





**Table 3:** All parameter value ranges for the suite of sensitivity simulations ran in TOMAS.

| Parameter (abbreviation) | Base value [unit] | Multipliers |
|---|---|---|
| Nucleation rate constant ($k_{nuc}$) | $1 \times 10^{-21}$ [cm$^{-6}$ s$^{-1}$] | 0.05, 0.1, 0.25, 0.5, 1, 2, 4, 10, 20 |
| OH oxidation rate constant ($k_{OH}$) | $k_{OH} = -5.7 \times 10^{-12} \ln(C^*) + 1.14 \times 10^{-10}$ [cm$^3$ molec$^{-1}$ s$^{-1}$] | 0.1, 0.2, 0.4, 0.7, 1, 1.5, 2.5, 5, 10 |
| Reactive uptake coefficient ($\gamma_{OH}$) | 0.6 [unitless] | 0.25, 0.5, 1, 2, 4 |
| Effective uptake coefficient ($\alpha_{EFF}$) | 1 [unitless] | 0.01, 0.05, 0.1, 1 |
| Gas-phase fragmentation rate constant ($k_{ELVOC}$) | $1 \times 10^{-10}$ [cm$^3$ s$^{-1}$] | 0.11, 0.33, 1, 3, 9 |



**Table 4:** Modelled fractional contribution of initial S/IVOCs towards the total change in number and volume between the initial and final number volume size distributions of each exposure modelled in this study. We use the measured S/IVOCs for the BEACHON-RoMBAS calculations and the best-fit initial S/IVOC concentration found for the GoAmazon calculations. The remaining fractional contribution towards the total change in number and volume is attributable to the measured initial monoterpenes (both campaigns) and measured initial isoprene (GoAmazon). Each exposure's fractional contribution is calculated using the averaged contributions of the 6 model cases with the lowest (best) NME values from the full model parameter space.

| Exposure (eq. age) | Fractional contribution from S/IVOCs (number) | Fractional contribution from S/IVOCs (volume) |
|---|---|---|
| BEACHON-RoMBAS | | |
| 0.090 | 0.89 | 0.20 |
| 0.098 | 0.86 | 0.05 |
| 0.16 | 1.0 | 0.29 |
| 0.23 | 0.79 | 0.66 |
| 0.27 | 0.93 | 0.68 |
| 0.77 | 0.55 | 0.55 |
| 0.82 | 0.94 | 0.66 |
| 0.91 | 0.89 | 0.64 |
| Average | 0.85 | 0.39 |
| GoAmazon | | |
| 0.39 | 1.0 | 0.35 |
| 0.40 | 1.0 | 0.42 |
| 0.51 | 1.0 | 0.71 |
| 0.53 | 1.0 | 0.76 |
| Average | 1.0 | 0.66 |





**Table 5:** Modelled fractional contribution of the measured initial isoprene concentrations towards the total change in number and volume between the initial and final number and volume size distributions modelled from the GoAmazon2014/5 campaign, using the best-fit S/IVOC estimate. (Isoprene was not included in the model for the BEACHON-RoMBAS
5    distributions.) The remaining fractional contribution is attributable to the MT and S/IVOC concentrations in the model. Each exposure's fractional contribution is calculated using the averaged contributions of the six model cases with the lowest (best) NME values from the full model parameter space.

| Exposure (eq. age) | Fractional contribution from isoprene (number) | Fractional contribution from isoprene (volume) |
|---|---|---|
| GoAmazon | | |
| 0.39 | 0.0 | 0.0 |
| 0.40 | 0.02 | 0.0 |
| 0.51 | 0.0 | 0.0 |
| 0.53 | 0.0 | 0.03 |
| Average | 0.0 | 0.01 |





**Figures**

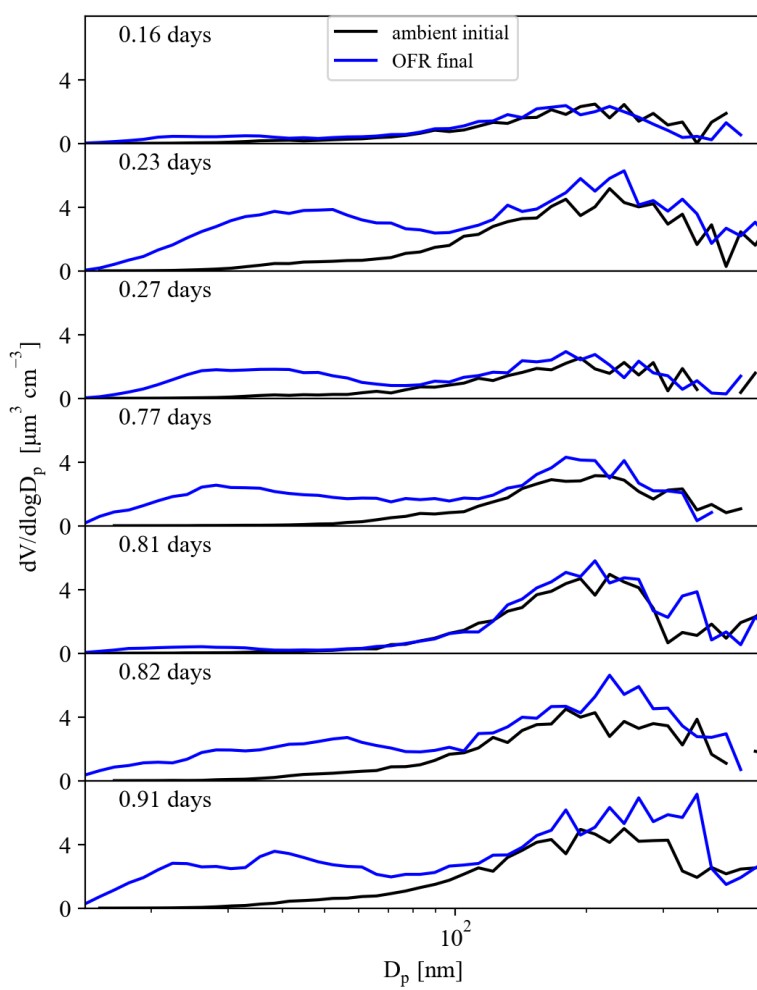

5 **Figure 1.** BEACHON-RoMBAS initial (i.e. ambient air, black line) and final (i.e. after OFR processing, blue line) SMPS-derived volume distributions for each individual exposure modelled in this study. The differences in SOA production between exposures of similar ages are due to the fact that the exposures were taken from different times during the campaign and thus different precursor concentrations were present (Table 2).





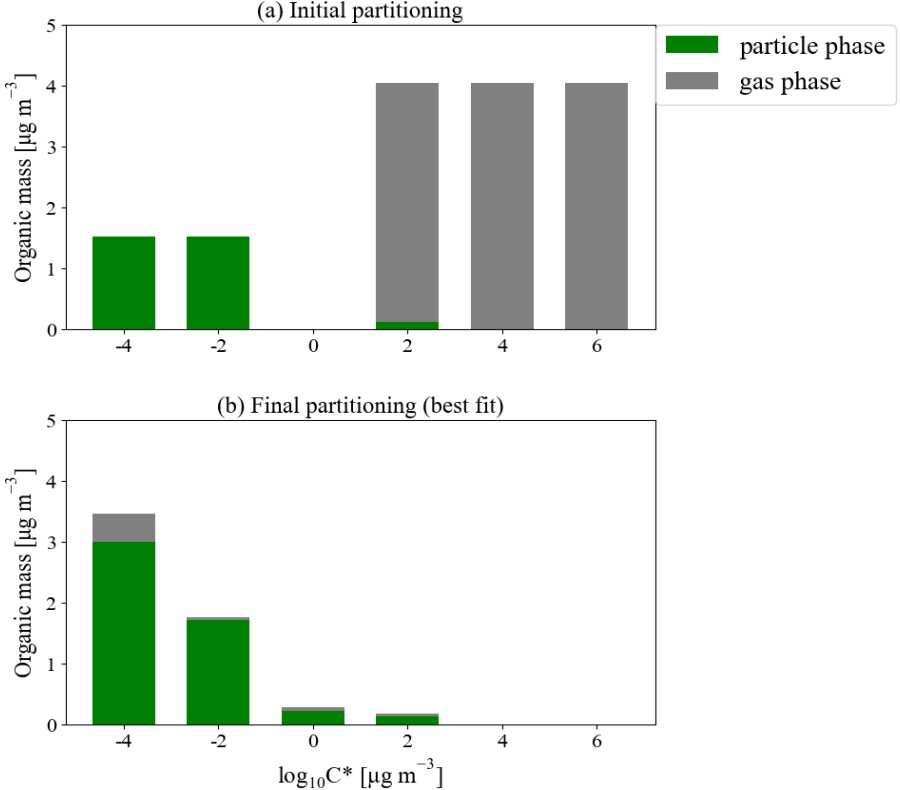

**Figure 2.** Example model (a) initial ambient and (b) final modelled partitioning for a 0.23 eq. day aging exposure from the BEACHON-RoMBAS campaign, with the particle phase loadings in green and gas phase loadings in grey (all in $\mu$g m$^{-3}$).

5   The initial S/IVOC concentration is evenly divided between the C*=10$^2$ to C*=10$^6$ $\mu$g m$^{-3}$ bins; the initial total aerosol mass is evenly divided between the C*=10$^{-4}$ to C*=10$^{-2}$ $\mu$g m$^{-3}$ bins. The C*=10$^0$ $\mu$g m$^{-3}$ bin is assumed to have an initial concentration of 0 $\mu$g m$^{-3}$. The input VOCs (MT for BEACHON-RoMBAS and MT and isoprene for GoAmazon2014/5) are assumed to be in a volatility bin greater than the C*=10$^6$ $\mu$g m$^{-3}$ bin (not shown). Panel (b) is the best fit modelled final partitioning for this exposure, corresponding to 2×k$_{NUC1}$, 5×k$_{OH}$, 0.5× $\gamma_{OH}$, k$_{ELVOC}$, and $\alpha_{EFF}$ = 0.01. The C*=10$^{-4}$ $\mu$g m$^{-3}$ bin

10   (assumed to represent ELVOCs) shows a significant amount of material remaining in the gas phase at the end of the modelled exposure, indicating that the production of gas-phase ELVOCs exceeded the timescale of condensation and gas-phase fragmentation within in the OFR.





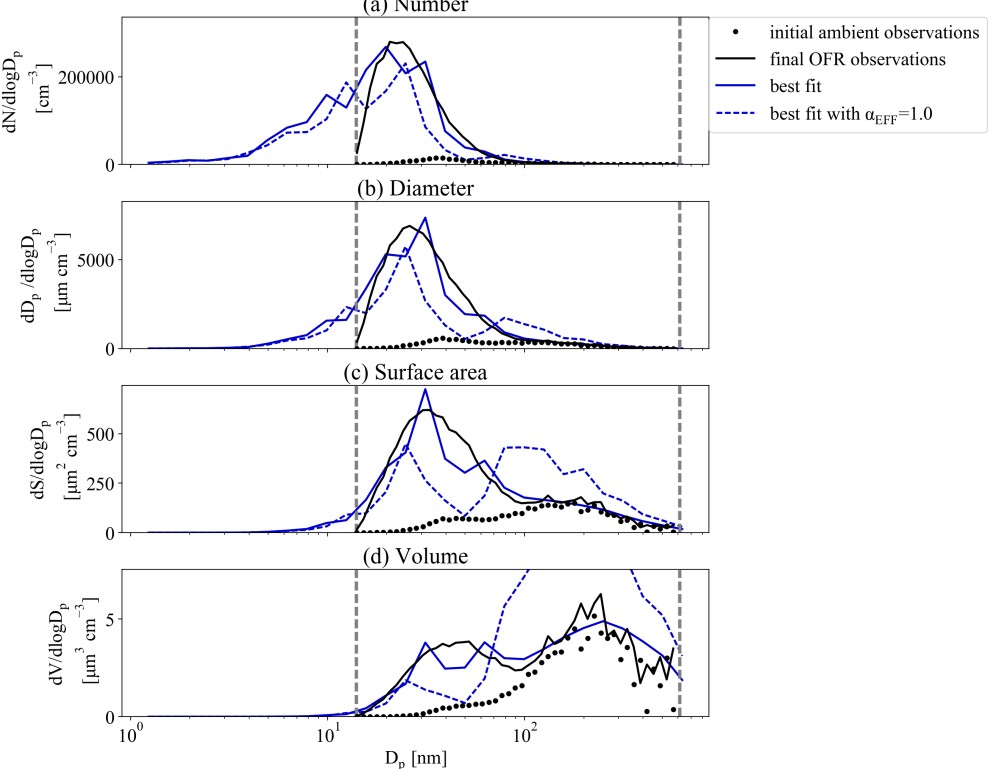

**Figure 3.** Example case of a 0.23 eq. day aging exposure from the BEACHON-RoMBAS campaign. The panels represent the moments used to calculate the normalized mean error (NME), with (a) as particle number, (b) as particle diameter (also referred to as aerosol length), (c) as particle surface area, and (d) as particle volume. The NME is calculated for each model run, using the final (OFR output) observed size distribution (black lines) compared to each model run's final size distribution (colored lines). The solid blue lines are for the best-fit model case for this exposure, corresponding to $2 \times k_{NUC1}$, $5 \times k_{OH}$, $0.5 \times \gamma_{OH}$, $k_{ELVOC}$, and $\alpha_{EFF} = 0.01$ (NME = 0.03). The dashed orange lines are for the same parameter values of the best-fit case except that $\alpha_{EFF} = 1.0$ (NME = 0.3). The vertical grey dashed lines indicate the particle size range across which the integration for calculating each mean moment was computed. The initial observed ambient size distribution (dotted black lines) is also plotted for comparison.





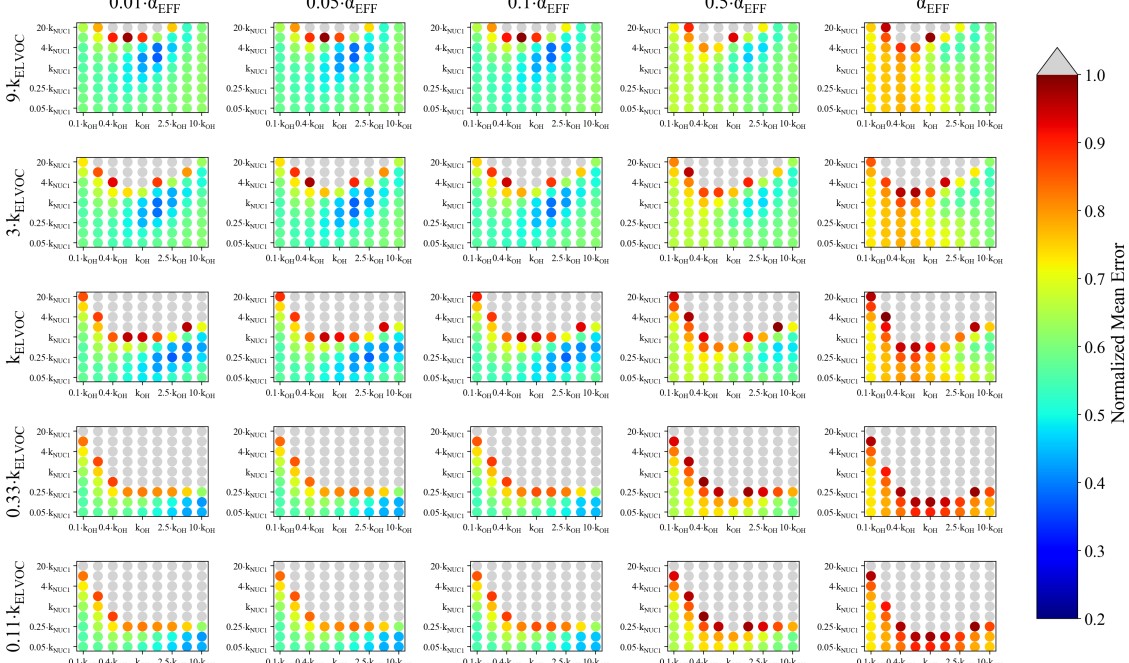

**Figure 4.** Representation of the parameter space for the average across the 0.09-0.9 day eq. aging exposures from BEACHON-RoMBAS examined in this study for the NUC1 nucleation scheme and base value of the reactive uptake coefficient of 0.6. The effective accommodation coefficient increases across each row of panels; the rate constant of gas-phase fragmentation increases up each column of panels. Within each panel, the rate constant of gas-phase reactions with OH increases along the x-axis and the rate constant for nucleation increases along the y-axis. The color bar indicates the normalized mean error (NME) value for each simulation, with the lowest values indicating the least error between model and measurement. Grey regions indicate regions within the parameter space whose NME value is greater than 1. No averaged case had a NME value less than 0.2 for the cases shown here.





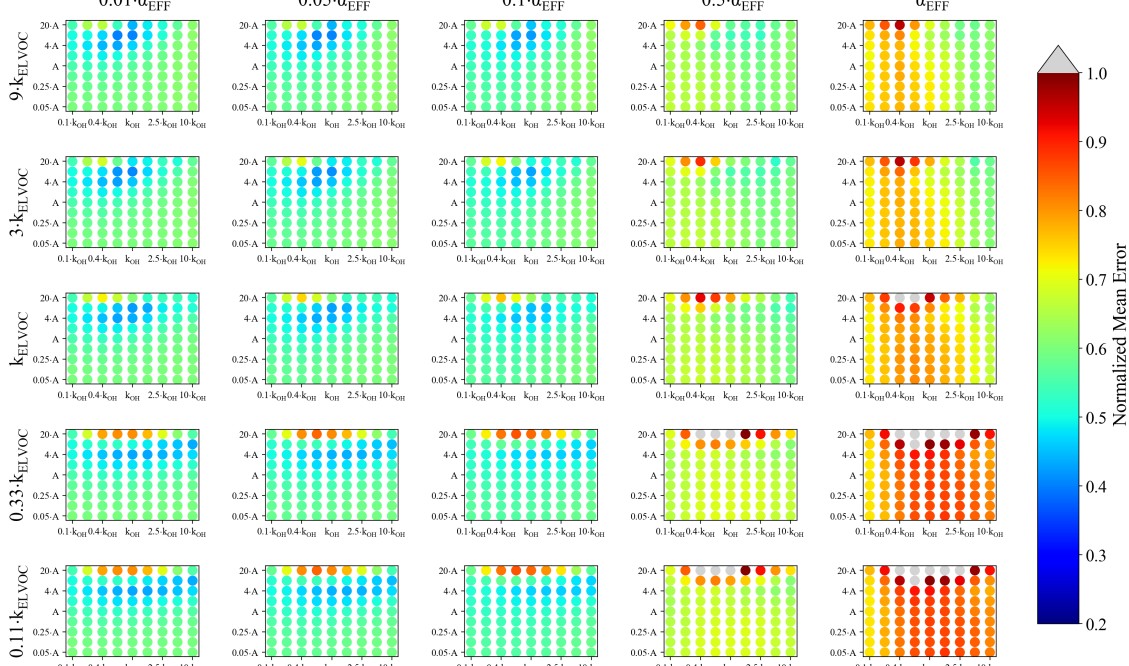

**Figure 5.** Representation of the parameter space for the average across the 0.09-0.9 day eq. aging exposures from BEACHON-RoMBAS examined in this study for the ACT nucleation scheme and base value of the reactive uptake coefficient of 0.6. The effective accommodation coefficient increases across each row of panels; the rate constant of gas-phase fragmentation increases up each column of panels. Within each panel, the rate constant of gas-phase reactions with OH increases along the x-axis and the rate constant for nucleation increases along the y-axis. The color bar indicates the normalized mean error (NME) value for each simulation, with the lowest values indicating the least error between model and measurement. Grey regions indicate regions within the parameter space whose NME value is greater than 1. No averaged case had a NME value less than 0.2 for the cases shown here.



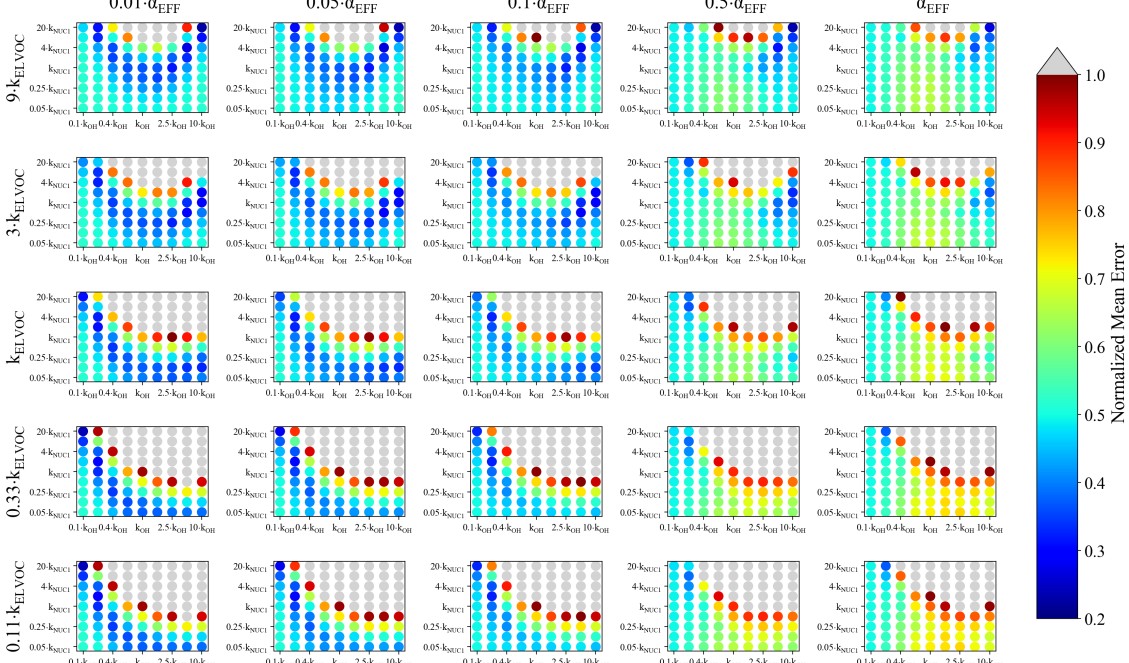

**Figure 6.** Representation of the parameter space for the average across the 0.3-0.6 day eq. aging exposures from GoAmazon examined in this study for the NUC1 nucleation scheme, base value of the reactive uptake coefficient of 0.6, and assumed S/IVOC:MT ratio of 30 times that of the BEACHON-RoMBAS S/IVOC:MT ratio. The effective accommodation coefficient increases across each row of panels; the rate constant of gas-phase fragmentation increases up each column of panels. Within each panel, the rate constant of gas-phase reactions with OH increases along the x-axis and the rate constant for nucleation increases along the y-axis. The color bar indicates the normalized mean error (NME) value for each simulation, with the lowest values indicating the least error between model and measurement. Grey regions indicate regions within the parameter space whose NME value is greater than 1. No averaged case had a NME value less than 0.2 for the cases shown here.