# Peer review of "Constraining nucleation, condensation, and chemistry in oxidation flow reactors using size-distribution measurements and aerosol microphysical modelling"

_Atmospheric Chemistry and Physics, 2018_

## Referee Comment (RC1) · Anonymous Referee #1 · 13 Jun 2018

The manuscript 'Constraining nucleation, condensation, and chemistry in oxidation flow reactors using size-distribution measurements and aerosol microphysical modelling' by Anna Hodshire and co-workers presents a very detailed description of chemical and physical properties and processes which have to be taken into account when modelling the size distribution evolution of ambient air after applying an oxidation flow reactor. The authors apply data from two intensive and well know field campaigns (GoAmazon2014/15 and BEACHON-RoMBAS) in the TwO-Moment Aerosol Sectional microphysics zero-dimensional model TOMAS. The description of the applied setup and the

uncertainties in several processes are well discussed. However, this is a very complex topic with significant impact for further research as OFRs applications in aerosol research increased in the last years. The main outcome of the manuscript is an overview about the probability of 5 selected microphysical processes which are crucial for the evolution of the size distribution under oxidative aging in the OFRs.

Personally I have to say that it was a pleasure and also very interesting to go through all the discussions on the different processes which were well provided with an adequate literature study. The authors also pointed clear out which processes were not considered and why and discussed the weakness of OFRs - so no need to discuss this further here in the review. The authors (and I believe this is more related to the first-author) performed a very large number of simulations on the topic and analysed the outcomes in a sufficient manner. They also mentioned and explained why several processes could not be considered and taking the already immense amount of performed simulations into account it is acceptable that this would be out of the frame of this work but was considered for future studies. I would see the work by Hodshire and co-workers as an important starting point on this topic and the paper will serve many other scientists as a look-up table in their future research. I have only some minor comments to the manuscript and would otherwise recommend that this paper gets published in ACP in the way it is without additional scientific improvements.

Minor comments for consideration:

Page 9 line 10-25: This paragraph is very difficult to understand and also I was reading it 3 times I still don't get all the values correct. So please rewrite it and I would also suggest to make it more clear in a table.

Page 15 line 18: . . . factors explore . . . should be . . . factors explored . . .

Page 25 line 30: . . . assumed decreased . . . should be . . . assumed decrease . . .

Page 50 line 8: The word "orange lines" should be replaced by blue lines in the text or

they should be changed to orange in the picture. And consequently this should then be done on page 19 line 28.

---

## Referee Comment (RC2) · Anonymous Referee #2 · 21 Jun 2018

This is a nice modeling study that utilizes measurements of SOA formation potential of ambient air organic mixtures at two separate field locations. The study demonstrates the utility of the OFR combined with a model to understand processes representing oxidation of organic gases, new particle formation, and growth.

I have the following suggestions and questions that need to be addressed before the Manuscript is accepted: The Manuscript would benefit if there are specific discussions about what OFR processes/parameters could be relevant in the atmosphere, and which of these are less applicable. For example: 1. The OFR does not represent high NOx

conditions. But the reality is that NOx is necessary for oxidant cycling. Granted that the OFR by design creates high OH concentrations even at low NOx. This is fine for reacting carbon. But the product and species distributions created this way in the OFR could be very different than those occurring due to NOx-mediated oxidant cycling in the atmosphere, even if the oxidant concentrations are the same. Please provide some discussions along these lines.

2. Due to the same reasoning as (1), please comment on whether the OFR can be used to study actual anthropogenic-biogenic interactions in the atmosphere. Note these interactions are NOx dependent.

3. Due to fast OFR processes, there is significant amount of ELVOC remaining in the gas-phase e.g. in Figure 2b. The authors explain that this is because the production of gas-phase ELVOCs exceeds the timescale of condensation and gas-phase fragmentation in the OFR. But in the real atmosphere, this will not be true since there is enough time for condensation and gas-phase fragmentation. Can the authors throw some light on this OFR-atmosphere timescale difference through any of their sensitivity studies?

4. The authors only consider fragmentation of the lowest ELVOC bin. On page 9, the authors mention fragmentation leads to more non-volatile products. It seems this is an error in the description. Fragmentation should lead to more volatile products. Please correct.

5. The argument that high kELVOC in the ELVOC bin effectively accounts for lack of fragmentation in the higher volatility bins, is not convincing. The mass of vapors in the higher volatility bins is much higher than the ELVOC bin. Also how fragmentation in higher volatility bins affects NP depends on details of oxidation, movement of species across the volatility intervals, the addition of functional groups, and particle phase processes (e.g. diffusion limitations etc.). So I find this statement as a major oversimplification. Please reframe this as a sensitivity study instead.

6. The fact that SIVOCs contribute so much to SOA potential over the Amazon seems

a bit weird. The rainforest is dominated by biogenic VOCs. Is this conclusion only valid for the dry season (where biomass burning is high) and not so much for the wet season?

7. What is the role of SIVOCs from biomass burning in the SOA formation potential over the Amazon?

---

## Author Comment (AC1) · 17 Jul 2018

We thank the reviewers for their comments on our paper. To guide the review process we have copied the reviewer comments in black text. Our responses are in regular blue font. We have responded to all the referee comments and have made the following alterations to our paper and supporting information (**in bold text**).

Before we address the specific comments, while revising the paper we have caught an error in our model and text that we address here and in the revised paper text and supplemental information:

Eq. (1) in the main text for the rate constant of vapor reactions with OH ($k_{OH}$) is from Jathar et al. (2014) lists that:

$$k_{OH} = -5.7 \times 10^{-12} \ln(C^*) + 1.14 \times 10^{-10} \quad \text{(a)}$$

This is the correct equation; however in the original Jathar et al. (2014) publication, the equation was listed as:

$$k_{OH} = -5.7 \times 10^{-12} \log(C^*) + 1.14 \times 10^{-10} \quad \text{(b)}$$

Although that 'log' term in (b) should be interpreted as 'ln' (Shantanu Jathar, personal communication), it was mistakenly written as 'log10' in our model. We have rerun the BEACHON-RoMBAS cases, using the first nucleation scheme ('NUC1') to reproduce the representation of the parameter space for the average across the 0.09-0.9 day eq. aging exposures as shown in Fig. 4. We found that the new $k_{OH}$ values using ln($C^*$) do not fit the averaged observations as well. As well, a factor of 10 division for the nucleation rate values is required to match the shape of the original model fits; even so, the fit is not as good as was for log10($C^*$). This indicates that more investigation is required to better-constrain the functionalization rate constants of air containing a mixture of species. We provide the following changes, discussions, and figures in the main text and in the SI, section S5. (Note that many of the SI figure numbers are updated in the main text and SI; we do not show that here for brevity.)

(Main text)
(page 9 sect. 2.3.1) **In this study, gas-phase functionalization is modelled by assuming that the organic compounds within the VBS bins react with OH and products from this reaction drop by one volatility bin (a factor of 100 drop in volatility). As a base assumption of the rate constants of our vapors in the VBS bins reacting with OH ($k_{OH}$), we use the relationship developed for aromatics by Jathar et al. (2014), based on data from Atkinson and Arey (2003a):**

$$k_{OH} = -5.7 \times 10^{-12} log10(C*) + 1.14 \times 10^{-10} \qquad \textbf{(7)}$$

As the assumption that the ambient mixture of S/IVOCs is similar to those of aromatics may not be suitable, we treat the rate constants for this volatility-reactivity relationship as an uncertain parameter that we vary in this study (Section 2.3.3). Further, it has been realized after the initial completion of this study that the first term in Eq. 1 is instead $-5.7 \times 10^{-12} ln(C*)$ (S. Jathar, personal communication). We discuss the differences and implications in using $log10(C*)$ versus $ln(C*)$ in Sect. 3.1.1.

(page 15, Sect. 2.3.3) As discussed in Sect. 2.3.1, we use as the base rate of $k_{OH}$ the relationship determined for aromatics by Jathar et al. (2014), Eq. 1. (Again, we note that although we use $log10(C*)$ in the first term of Eq. 1, $ln(C*)$ is the correct expression for the fit found in Jather et al., 2014; S. Jathar, personal communication).

(page 21, Sect. 3.1.1) As discussed in Sect. 2.3.1, the first term of Eq. 1 relies on $log10(C*)$ for the rate constant of $k_{OH}$; however, the fit of Jathar et al. (2014) should instead use $ln(C*)$:

$$k_{OH} = -5.7 \times 10^{-12} ln(C*) + 1.14 \times 10^{-10} \qquad (7)$$

(S. Jathar, personal communication). Table S1 gives the numerical results for $k_{OH}$ for both Eq 1. and Eq. 7; when Eq. 7 is used, the highest volatility bin reacts ~2 times more quickly but the rate constants converge for $C*=10^0$ µg m$^{-3}$ and remain similar to each other for the lowest volatility bins. Figures S21 and S22 provide results of the parameter space for the average across the 0.09-0.9 day eq. aging exposures from BEACHON-RoMBAS examined in this study, using the NUC1 nucleation scheme and base value of the reactive uptake coefficient of 0.6, using Eq. 7 for $k_{OH}$ (using the same multipliers for $k_{OH}$ as listed in Table 3). Figure S21 uses all parameter values listed in Table 3 (excepting the updated $k_{OH}$ values) and can be directly compared to Fig. 4. Figure S22 further decreases each nucleation rate constant ($k_{NUC1}$) value by a factor of 10 in order to match the shapes of each panel of Fig. 4. Although Fig. S22 well-matches the general shapes seen in Fig. 4 for each $k_{ELVOC}$ and $\alpha_{EFF}$, the normalized mean errors are larger in both Figs. S21 and S22 than in Fig. 4. Thus we conclude that for this study, using the $k_{OH}$ values from Eq. 1 provide better fits and that parameterizations for rate constants for $k_{OH}$ of air containing a mixture of ambient species require further investigation.

(page 28, Sect. 4) Similarly, we found that gas-phase oxidation rate constants similar to that of Jathar et al. (2014), fit from aromatics, allowed for good fits (we assumed that these reactions were 100% functionalization and treated the fragmentation reactions separately).

The gas-phase oxidation rate constants provided better fits when using a slightly different formulation than the parameterization from Jathar et al. (2014), indicating that further studies are required for fitting parameterizations for air containing a mixture of ambient species.

(Supplement additions)

**S5. Sensitivity of the model to the $k_{OH}$ formulation**
All model runs in this paper have been performed using Eq. 1 in the main text for the gas-phase functionalization rate constant between organic vapors and OH:
$$k_{OH} = -5.7 \times 10^{-12} log10(C*) + 1.14 \times 10^{-10} \qquad \text{(S1)}$$
This equation is from Jathar et al. (2014); however, the equation should instead be
$$k_{OH} = -5.7 \times 10^{-12} ln10(C*) + 1.14 \times 10^{-10} \qquad \text{(S2)}$$
(S. Jathar, personal communication). Table S1 provides the $k_{OH}$ values obtained from Eqs. S1 and S2. Figure S21 is a direct comparison to Fig. 4 of the main text, showing the results of the parameter space for the average across the 0.09-0.9 day eq. aging exposures from BEACHON-RoMBAS examined in this study, using the NUC1 nucleation scheme and base value of the reactive uptake coefficient of 0.6, and the $k_{OH}$ formulation of Eq. S2, keeping all other parameter values identical to the values listed in Table 3. (We still test the same multipliers on $k_{OH}$ listed in Table 3). Figure S22 provides the same figure as Fig. S21, but with the nucleation rate values ($k_{NUC1}$) each decreased by a factor of 10 from that of the values in Table 3. Although Fig. S22 well-matches the general shapes seen in Fig. 4 for each $k_{ELVOC}$ and $\alpha_{EFF}$, the normalized mean errors are larger in both Figs. S21 and S22 than in Fig. 4. Thus we conclude that for this study, using the $k_{OH}$ values from Eq. S1 provide better fits and that parameterizations for rate constants for $k_{OH}$ of air containing a mixture of ambient species require further investigation.

**Table S1: $k_{OH}$ values obtained from Eq. S1 (Eq. 1 of the main text) and Eq. S2 for each volatility bin used in this study.**

| $C*$ [$\mu$g m$^{-3}$] | $k_{OH}$ from Eq. S1 [cm molec$^{-1}$ s$^{-1}$] | $k_{OH}$ from Eq. S2 [cm$^3$ molec$^{-1}$ s$^{-1}$] |
|---|---|---|
| $1 \times 10^{-4}$ | $1.36 \times 10^{-10}$ | $1.66 \times 10^{-10}$ |
| $1 \times 10^{-2}$ | $1.24 \times 10^{-10}$ | $1.40 \times 10^{-10}$ |
| $1 \times 10^{0}$ | $1.14 \times 10^{-10}$ | $1.14 \times 10^{-10}$ |
| $1 \times 10^{2}$ | $1.03 \times 10^{-10}$ | $8.78 \times 10^{-11}$ |
| $1 \times 10^{4}$ | $9.12 \times 10^{-11}$ | $6.15 \times 10^{-11}$ |
| $1 \times 10^{6}$ | $7.98 \times 10^{-11}$ | $3.53 \times 10^{-11}$ |

[Figure]

**Figure S21. Representation of the parameter space for the average across the 0.09-0.9 day eq. aging exposures from BEACHON-RoMBAS examined in this study for the NUC1 nucleation scheme and base value of the reactive uptake coefficient of 0.6, using Eq. S2 for the values of $k_{OH}$. The effective accommodation coefficient increases across each row of panels; the rate constant of gas-phase fragmentation increases up each column of panels. Within each panel, the rate constant of gas-phase reactions with OH increases along the x-axis and the rate constant for nucleation increases along the y-axis. The color bar indicates the normalized mean error (NME) value for each simulation, with the lowest values indicating the least error between model and measurement. Grey regions indicate regions within the parameter space whose NME value is greater than 1. No averaged case had a NME value less than 0.2 for the cases shown here. This figure can be directly compared to Fig. 4 of the main text.**

[Figure]

**Figure S22. Representation of the parameter space for the average across the 0.09-0.9 day eq. aging exposures from BEACHON-RoMBAS examined in this study for the NUC1 nucleation scheme and base value of the reactive uptake coefficient of 0.6, using Eq. S2 for the values of $k_{OH}$ and dividing each original $k_{NUC1}$ value from Table 3 by a factor of 10. The effective accommodation coefficient increases across each row of panels; the rate constant of gas-phase fragmentation increases up each column of panels. Within each panel, the rate constant of gas-phase reactions with OH increases along the x-axis and the rate constant for nucleation increases along the y-axis. The color bar indicates the normalized mean error (NME) value for each simulation, with the lowest values indicating the least error between model and measurement. Grey regions indicate regions within the parameter space whose NME value is greater than 1. No averaged case had a NME value less than 0.2 for the cases shown here.**

**Anonymous Referee #1**

Overview

The manuscript 'Constraining nucleation, condensation, and chemistry in oxidation flow reactors using size-distribution measurements and aerosol microphysical modelling' by Anna Hodshire and co-workers presents a very detailed description of chemical and physical properties and processes which have to be taken into account when modelling the size distribution evolution of ambient air after applying an oxidation flow reactor. The authors apply data from two intensive and well know field campaigns (GoAmazon2014/15 and BEACHON-RoMBAS) in the TwO-Moment Aerosol Sectional microphysics zero-dimensional model TOMAS. The description of the applied setup and the uncertainties in several processes are well discussed. However, this is a very complex topic with significant impact for further research as OFRs applications in aerosol research increased in the last years. The main outcome of the manuscript is an overview about the probability of 5 selected microphysical processes which are crucial for the evolution of the size distribution under oxidative aging in the OFRs. Personally I have to say that it was a pleasure and also very interesting to go through all the discussions on the different processes which were well provided with an adequate literature study. The authors also pointed clear out which processes were not considered and why and discussed the weakness of OFRs - so no need to discuss this further here in the review. The authors (and I believe this is more related to the first-author) performed a very large number of simulations on the topic and analysed the outcomes in a sufficient manner. They also mentioned and explained why several processes could not be considered and taking the already immense amount of performed simulations into account it is acceptable that this would be out of the frame of this work but was considered for future studies. I would see the work by Hodshire and co-workers as an important starting point on this topic and the paper will serve many other scientists as a look-up table in their future research. I have only some minor comments to the manuscript and would otherwise recommend that this paper gets published in ACP in the way it is without additional scientific improvements.

Minor comments for consideration:

R1.1) Page 9 line 10-25: This paragraph is very difficult to understand and also I was reading it 3 times I still don't get all the values correct. So please rewrite it and I would also suggest to make it more clear in a table.

We agree that this paragraph is confusing as originally written. We've updated as follows:

**We further account for monoterpenes (MT) oxidation by OH for both campaigns and isoprene oxidation by OH for GoAmazon in the model. Palm et al. (2016) determined that on average during the BEACHON campaign, MT contributed 20% of the measured SOA formation, with sesquiterpenes (SQT), isoprene, and toluene contributing an additional 3% of the measured SOA formation. Since these other VOCs contributed a minor amount to the measured SOA formation, they were not included in this analysis. S/IVOCs at BEACHON contributed the remaining 77% towards the measured SOA formation, and were likely the main source for new particles in the OFR. It was observed that for the GoAmazon campaign during the dry season, the approximate average contribution to the measured SOA was 4% from isoprene and 4% from MT, with an 8% remaining contribution towards the measured SOA coming from SQT, benzene, toluene, xylenes, and trimethylbenzene (TMB), combined. Thus, less of the total SOA can be described by the VOCs included in the model (isoprene and MT) for the GoAmazon simulations than can be described for the BEACHON campaign. The remaining 83% of measured SOA formation was found to have come from unmeasured S/IVOCs, so again S/IVOCs were likely the main source for new particles in the OFR. Including the other VOCs would only increase the model-predicted SOA yield from the initial VOCs by a few tenths of a $\mu g\ m^3$, and decrease the model-predicted SOA yield from the initial S/IVOCs by a similar amount, and so they were excluded for simplicity.**

We hope that this text is now clear enough that a table is not necessary.

R1.2) Page 15 line 18: . . . factors explore . . . should be . . . factors explored . . .

We've modified the text.

R1.3) Page 25 line 30: . . . assumed decreased . . . should be . . . assumed decrease . . .

We've modified the text.

R1.4) Page 50 line 8: The word "orange lines" should be replaced by blue lines in the text or they should be changed to orange in the picture. And consequently this should then be done on page 19 line 28.

Good catch--we've modified the text and figure caption to read as "blue lines".

**Anonymous Referee #2**

Overview

This is a nice modeling study that utilizes measurements of SOA formation potential of ambient air organic mixtures at two separate field locations. The study demonstrates the utility of the OFR combined with a model to understand processes representing oxidation of organic gases, new particle formation, and growth. I have the following suggestions and questions that need to be addressed before the Manuscript is accepted: The Manuscript would benefit if there are specific discussions about what OFR processes/parameters could be relevant in the atmosphere, and which of these are less applicable. For example:

R2.1) The OFR does not represent high NOx conditions. But the reality is that NOx is necessary for oxidant cycling. Granted that the OFR by design creates high OH concentrations even at low NOx. This is fine for reacting carbon. But the product and species distributions created this way in the OFR could be very different than those occurring due to NOx-mediated oxidant cycling in the atmosphere, even if the oxidant concentrations are the same. Please provide some discussions along these lines.

Indeed in the atmosphere, NO and $NO_2$ are critical to HOx radical cycling. In the OFR as operated in these studies, NO is very low due to the high levels of $O_3$ and OH (Li et al., 2015; Peng et al., 2017). However, the use of the UV lamps at 185/254 nm allows sustaining the radical cycling even under low NO levels (Li et al., 2015; Peng et al., 2015). As long as the radical composition is similar to the atmosphere, there is no reason for the product distributions to be different for this reason.

Perhaps what the reviewer is trying to say is that in the atmosphere some organic peroxy radicals ($RO_2$) will react with NO, while this is not the case in the OFR as operated here. We have modified / added the following text to section 2.1 of the Methods to address this point:

(page 7) **The chemical regime was relevant to ambient OH oxidation, as discussed in detail in Peng et al. (2015, 2016). We note that about ~½ of the $RO_2$ radicals reacted with NO in ambient air during BEACHON-RoMBAS (Fry et al., 2013), but this was not the case in the OFR due to very rapid oxidation of NO (Li et al., 2015; Peng et al., 2017). Thus some differences in the product distributions for ambient vs. OFR oxidation would be expected. Recently, new OFR methods have been developed that allow $RO_2$+NO to dominate (Lambe et al., 2017; Peng et al., 2018), but those methods were not available at the time of the field studies discussed here.**

R2.2) Due to the same reasoning as (1), please comment on whether the OFR can be used to study actual anthropogenic-biogenic interactions in the atmosphere. Note these interactions are NOx dependent.

"Anthropogenic-biogenic interactions" is a very broad topic that can mean different things to different researchers. For example the impact of anthropogenic $SO_2$ on biogenic SOA formation is an example of an interaction that can and has been studied with an OFR (Friedman et al., 2016). The effect of $NO_x$ on biogenic SOA formation can also be studied with the newer OFR techniques (Lambe et al., 2017; Peng et al., 2018). Thus we see no reason why the OFR is any more challenged than any other technique to study many types of anthro-bio interactions, as long as the studies are designed carefully.

R2.3) Due to fast OFR processes, there is significant amount of ELVOC remaining in the gas-phase e.g. in Figure 2b. The authors explain that this is because the production of gas-phase ELVOCs exceeds the timescale of condensation and gas-phase fragmentation in the OFR. But in the real atmosphere, this will not be true since there is enough time for condensation and gas-phase fragmentation. Can the authors throw some light on this OFR-atmosphere timescale difference through any of their sensitivity studies?

We do briefly discuss the OFR vs. ambient timescale differences in our work, on page 23: "The OFR shifts the relative timescales of chemistry versus condensation, which may create higher concentrations of low-volatility vapors capable of participating in nucleation and early growth relative to the ambient atmosphere during GoAmazon." We see that for increasing eq. age exposures, more ELVOCs are in the particle phase than the gas phase. To further address the reviewer's comment, we added the following discussion:

(page 18, section 3.1.1) **It should be noted that more gas-phase ELVOCs are being formed than could condense during the timescales of the simulated exposures (Fig. 2b). As ELVOCs would be formed more slowly in the ambient atmosphere but with a similar condensational loss timescale, nucleation is expected to proceed faster in the OFR than the ambient atmosphere. This is a reason for the potential usefulness of this OFR technique, that nucleation from chemistry of species present in ambient air can be studied, even if nucleation would not be occurrent under ambient-only conditions.**

(page 28, section 4) **We found that the nucleation rate constants for the $H_2SO_4$-organics nucleation mechanism suggested by Riccobono et al. (2014) allowed for good models fits, with the caveats that the temperatures of both campaigns were higher than the experimental conditions of Riccobono et al. (2014) (4-12 K higher for BEACHON and 18-19 K higher for GoAmazon), and that the timescales upon which ELVOCs were formed and capable of participating in nucleation could be shorter than that of the ambient atmosphere.**

R2.4) The authors only consider fragmentation of the lowest ELVOC bin. On page 9, the authors mention fragmentation leads to more non-volatile products. It seems this is an error in the description. Fragmentation should lead to more volatile products. Please correct.

We believe the reviewer mis-read this sentence (page 9) that reads "We account for gas-phase fragmentation reactions separately by allowing one OH reaction with a molecule in the lowest volatility bin ($C^*=10^{-4}$ $\mu$g m$^{-3}$; assumed to be an ELVOC molecule) to lead to an irreversible fragmentation into non-condensable volatile products that are no longer tracked in the model." (Underlined for clarification)

R2.5) The argument that high kELVOC in the ELVOC bin effectively accounts for lack of fragmentation in the higher volatility bins, is not convincing. The mass of vapors in the higher volatility bins is much higher than the ELVOC bin. Also how fragmentation in higher volatility bins affects NP depends on details of oxidation, movement of species across the volatility intervals, the addition of functional groups, and particle phase processes (e.g. diffusion limitations etc.). So I find this statement as a major oversimplification. Please reframe this as a sensitivity study instead.

We modify the following portions of the text:

(page 9) **We only allow for fragmentation of species in our lowest volatility bin in order to limit the number of parameters in our study, but we acknowledge that this is a limitation of this study and should be considered as a sensitivity study for fragmentation.**

(page 28) **However, the fragmentation scheme used in this study should be viewed as a sensitivity study; the inclusion of a more-complex fragmentation scheme would have added more free parameters to our study and will be left to a future study.**

R2.6) The fact that SIVOCs contribute so much to SOA potential over the Amazon seems a bit weird. The rainforest is dominated by biogenic VOCs. Is this conclusion only valid for the dry season (where biomass burning is high) and not so much for the wet season?

The rainforests' primary emissions may be dominated by biogenic VOCs but it is becoming evident that previously unmeasured/uncharacterized S/IVOCs contribute a non-trivial amount towards SOA formation under OH oxidation (but not $NO_3$ or $O_3$ oxidation) in ambient forested locations (Palm et al., 2016; 2017; 2018). Primary VOCs are, by definition, too volatile to condense onto aerosol and must undergo oxidation to form lower-volatility products; as a portion of these secondary products appear to be in the intermediate to semi-volatile range and can form SOA upon additional oxidation (e.g. Hunter et al., 2017; Palm et al., 2016). This can explain why

S/IVOCs can then contribute so much to SOA production. This is, however, still very much an understudied question as to how much S/IVOCs contribute towards SOA, especially in different environments. This is discussed in detail in Palm et al. (2016) and Hunter et al. (2017), and thus we do not elaborate further on this topic on the revised paper.

Figure 13 (b) of Palm et al. (2018) provides their estimation of potential SOA formed for the wet season and the dry season in the Amazon. Although the dry season has more total SOA potential than the wet season, biomass burning contributes under 20% towards the total SOA for both seasons. Thus it seems unlikely that even if 100% of all biomass burning emissions are in the S/IVOC range that biomass burning alone is the cause of the high apparent contribution to SOA formation found in this work and Palm et al. (2018).

R2.7) What is the role of SIVOCs from biomass burning in the SOA formation potential over the Amazon?

See response to R2.6. Biomass burning is a relatively small contributor to SOA potential during GoAmazon (Fig. 13 of Palm et al., 2018). The relative contributions of biomass burning SIVOCs vs. VOCs to that potential have not been studied quantitatively and it would be difficult to do so with the available data.

[revised manuscript text omitted]

---

## Author Comment (AC3) · 17 Jul 2018

[revised manuscript text omitted]

Anna Lily Hodshire 7/6/2018 4:40 PM

Anna Lily Hodshire 7/6/2018 4:40 PM

Anna Lily Hodshire 7/6/2018 4:40 PM

Anna Lily Hodshire 7/6/2018 4:40 PM

Anna Lily Hodshire 7/6/2018 4:40 PM

All model runs in this paper have been performed using Eq. 1 in the main text for the gas-phase functionalization rate constant between organic vapors and OH:

$$k_{OH} = -5.7 \times 10^{-12} \log_{10}(C*) + 1.14 \times 10^{-10} \qquad \text{(S1)}$$

This equation is from Jathar et al. (2014); however, the equation should instead be

$$k_{OH} = -5.7 \times 10^{-12} \log_{10}(C*) + 1.14 \times 10^{-10} \qquad \text{(S2)}$$

(S. Jathar, personal communication). Table S1 provides the $k_{OH}$ values obtained from Eqs. S1 and S2. Figure S21 is a direct comparison to Fig. 4 of the main text, showing the results of the parameter space for the average across the 0.09-0.9 day eq. aging exposures from BEACHON-RoMBAS examined in this study, using the NUC1 nucleation scheme and base value of the reactive uptake coefficient of 0.6, and the $k_{OH}$ formulation of Eq. S2, keeping all other parameter values identical to the values listed in Table 3. (We still test the same multipliers on $k_{OH}$ listed in Table 3). Figure S22 provides the same figure as Fig. S21, but with the nucleation rate values ($k_{NUC1}$) each decreased by a factor of 10 from that of the values in Table 3. Although Fig. S22 well-matches the general shapes seen in Fig. 4 for each $k_{ELVOC}$ and $\alpha_{EFF}$, the normalized mean errors are larger in both Figs. S21 and S22 than in Fig. 4. Thus we conclude that for this study, using the $k_{OH}$ values from Eq. S1 provide better fits and that parameterizations for rate constants for $k_{OH}$ of air containing a mixture of ambient species require further investigation.

**Table S1:** $k_{OH}$ values obtained from Eq. S1 (Eq. 1 of the main text) and Eq. S2 for each volatility bin used in this study.

[revised manuscript text omitted]

**Anna Lily Hodshire 7/6/2018 4:47 PM**

---

## Author Response (AR2)

Response to Reviewer #2 (technical correction)

We thank Reviewer #2 for their additional comment:

There is just 1 point which needs to be clarified in their response to R2.6. The authors describe how primary VOCs are too volatile and so S/IVOCs are needed for SOA formation. But here, it needs to be acknowledged that the authors refer to S/IVOCs as a broad class of species from all classes: anthropogenic, biogenic and biomass burning. This is important enough to be also mentioned more explicitly in the Manuscript Discussions.

 We note that in Section 3.2 (first paragraph) we state:
 "As BEACHON was dominated by biogenic emissions (primarily MTs), but GoAmazon had major contributions from anthropogenic and biomass burning sources as well as various biogenic emissions (Palm et al., 2018), the larger S/IVOC is thought to be dominated by emissions and partially oxidized products from the two latter sources."

To better discuss this point, we have added the following in the conclusions section (additions in bold):
"Like Palm et al. (2016; 2018), our results indicate the importance of S/IVOCs towards aerosol growth in the OFR at both the BEACHON and GoAmazon campaigns. We find that S/IVOCs contribute on average 85% and 39% (BEACHON) and 100% and 66% (GoAmazon) towards the change in total number and volume, respectively, for the exposures modelled in this study. **There remains uncertainty in the sources of these S/IVOCs: they could be directly emitted or formed as oxidation products from both biogenic and anthropogenic sources for BEACHON (Palm et al., 2016) and from biogenic, anthropogenic, and biomass burning sources for GoAmazon (Palm et al., 2018). Further studies are required to better understand, speciate, and quantify S/IVOC sources.**"

Finally, we note the following additions to the data availability statement:
"All data shown in the figures pertaining to model results in this paper (including Supplement) are available upon request. **The TOMAS-VBS model code used in this paper is available at https://hdl.handle.net/10217/190133.**"

and the acknowledgements:
**"**This research was supported by the US Department of Energy's Atmospheric System Research, an Office of Science, Office of Biological and Environmental Research program, under Grant No. DE-SC0011780, by the U.S National Oceanic and Atmospheric Administration, an Office of Science, Office of Atmospheric Chemistry, Carbon Cycle, and Climate Program, under the cooperative agreement award #NA17OAR430001, **and by the #NA17OAR4310002 and the**

**U.S. National Science Foundation, Atmospheric Chemistry program, under Grant No. AGS-1559607 and AGS-1558966.”**